# *Bacillus subtilis* encodes a discrete flap endonuclease that cleaves RNA-DNA hybrids

**Frances Caroline Lowder, Lyle A. Simmons** *

Department of Molecular, Cellular, and Developmental Biology, University of Michigan, Ann Arbor, Michigan, United States of America

* lasimm@umich.edu

**Data Availability Statement:** The relevant data are within the manuscript and supporting information files.

**Funding:** This study was funded by National Institutes of Health award (R35GM131772) https://www.nih.gov to LAS. The funders had no role in

## Abstract

The current model for Okazaki fragment maturation in bacteria invokes RNA cleavage by RNase H, followed by strand displacement synthesis and 5′ RNA flap removal by DNA polymerase I (Pol I). RNA removal by Pol I is thought to occur through the 5′-3′ flap endo/exonuclease (FEN) domain, located in the N-terminus of the protein. In addition to Pol I, many bacteria encode a second, Pol I-independent FEN. The contribution of Pol I and Pol I-independent FENs to DNA replication and genome stability remains unclear. In this work we purified *Bacillus subtilis* Pol I and FEN, then assayed these proteins on a variety of RNA-DNA hybrid and DNA-only substrates. We found that FEN is far more active than Pol I on nicked double-flap, 5′ single flap, and nicked RNA-DNA hybrid substrates. We show that the 5′ nuclease activity of *B. subtilis* Pol I is feeble, even during DNA synthesis when a 5′ flapped substrate is formed modeling an Okazaki fragment intermediate. Examination of Pol I and FEN on DNA-only substrates shows that FEN is more active than Pol I on most substrates tested. Further experiments show that Δ*polA* phenotypes are completely rescued by expressing the C-terminal polymerase domain while expression of the N-terminal 5′ nuclease domain fails to complement Δ*polA*. Cells lacking FEN (Δ*fenA*) show a phenotype in conjunction with an RNase HIII defect, providing genetic evidence for the involvement of FEN in Okazaki fragment processing. With these results, we propose a model where cells remove RNA primers using FEN while upstream Okazaki fragments are extended through synthesis by Pol I. Our model resembles Okazaki fragment processing in eukaryotes, where Pol δ catalyzes strand displacement synthesis followed by 5′ flap cleavage using FEN-1. Together our work highlights the conservation of ordered steps for Okazaki fragment processing in cells ranging from bacteria to human.

## Author summary

Proteins with 5′ flap endo/exonuclease (FEN) activity provide an essential contribution to DNA replication and repair in all cellular life. In bacteria, DNA polymerase I is thought to be the central enzyme involved in Okazaki fragment processing, using its DNA polymerase and 5′ nuclease activities to generate and then remove the 5′ ssRNA segment of an Okazaki fragment. Many bacterial genomes encode a second, discrete FEN in addition to

study design, data collection and analysis, decision to publish, or preparation of the manuscript.

**Competing interests:** The authors have declared that no competing interests exist.

Pol I. We show that FEN is the primary 5′ nuclease used by *B. subtilis* for primer removal. FEN activity exceeds that of Pol I on most substrates, including several that mimic Okazaki fragment intermediates. Additionally, we provide genetic evidence showing that FEN is involved in Okazaki fragment processing and that it is the DNA polymerase domain of Pol I rather than its 5′ nuclease domain that is important *in vivo*. With our results, we propose a new model for Okazaki fragment processing in *B. subtilis*, which may be prevalent in a wider group of bacteria than previously appreciated.

## Introduction

RNA-DNA hybrids form during transcription and DNA replication in all cells and many viruses. While their formation is essential, persistence of RNA-DNA hybrids can lead to genome instability by causing nicks and double-stranded breaks when the 2′ OH on the ribose sugar reacts with the phosphodiester bond, forming a 2′, 3′ cyclic phosphate [1]. During normal growth, RNA-DNA hybrids occur in several different forms within cells. R-loops represent one type of RNA-DNA hybrid, occurring when newly transcribed mRNA remains base-paired with the complementary DNA strand following transcription [2,3]. RNA-DNA hybrids also occur when ribonucleotides are covalently joined to DNA. Sugar errors can occur when single ribonucleotide monophosphates (rNMPs) or stretches of rNMPs are covalently nested in DNA by replicative polymerases [4,5] or under stress conditions when error-prone synthesis occurs [6]. Covalent RNA-DNA hybrids also form when replicative polymerases extend the primase-synthesized RNA primers at the initiation of leading strand synthesis and for synthesis of each Okazaki fragment during lagging strand replication [7–9]. Due to the frequency of Okazaki fragment synthesis, RNA primers occur predominantly on the lagging strand, resulting in the incorporation of approximately 20,000 rNMPs per genome replication event for bacteria [10]. Because RNA-DNA hybrids can lead to genomic instability and disease, all cells enlist a wide array of proteins to recognize and resolve the different types of hybrids that form *in vivo* [11]. The ribonuclease H (RNase H) proteins are a well characterized group of enzymes involved in the repair of all types of RNA-DNA hybrids occurring in cells. RNase HI, HII, and HIII are all capable of cleaving substrates with stretches of four or more ribonucleotides, which includes the RNA in Okazaki fragments. RNase HI and HIII are also able to resolve R-loops, while RNase HII is responsible for cleaving at single ribonucleotide errors in genomic DNA [12,13].

Several studies have provided evidence that RNase Hs participate in the removal of RNA primers during Okazaki fragment maturation [12–14], however, studies in *Escherichia coli* identified DNA polymerase I (Pol I) as the major bacterial protein involved in this process [14,15]. Pol I is composed of three domains: a 5′-3′ polymerase, a 3′-5′ exonuclease, and a 5′-3′ nuclease [16,17]. The well-studied Klenow fragment consists of the 5′-3′ polymerase and 3′-5′ exonuclease located in the C-terminus of the protein and is responsible for DNA synthesis and error "proofreading", while the 5′-3′ nuclease domain is located at the N-terminus of Pol I [15–20]. Coordination of Pol I activities during lagging strand synthesis should allow Pol I to synthesize DNA from an upstream Okazaki fragment through the downstream RNA primer followed by primer removal. The remaining nick between adjacent DNA fragments would then be sealed by DNA ligase to complete repair [15,21]. Due to the importance of Pol I activities during DNA replication, Pol I is very well conserved across bacterial species [22], although many bacteria, including *Bacillus subtilis*, lack the catalytic residues associated with an active 3′-5′ exonuclease [23], suggesting that the polymerase activity and 5′ nuclease activity of Pol I

are the most critical *in vivo* [24]. The current model, that Okazaki fragment maturation in bacteria is primarily accomplished by Pol I [15], stems from studies in *E. coli* and *B. subtilis* showing the accumulation of ribonucleotides in short DNA fragments from cells expressing various mutants of *polA* [25–28]. Based on these early studies, the overarching model for Okazaki fragment maturation in bacteria invokes RNA incision by RNase HI (or HIII) with DNA synthesis and the bulk of RNA removal catalyzed by Pol I.

More recent studies have used sequence and structural homology to identify the N-terminal nuclease domain of Pol I as part of a larger superfamily of proteins with 5′-3′ flap endonuclease and 5′-3′ exonuclease activity, known as FENs [29]. FEN activity is essential for replication; as such, proteins in this family are found across all domains of life [30–37]. Notable features of the FEN family include the binding of two or three divalent cations in the active site, a helical or unstructured arch near the active site, and a C-terminal DNA binding motif [30,38–41]. FENs are structure-specific nucleases best known for their endonuclease activity on substrates generated during strand-displacement synthesis such as 5′ flap overhang substrates or double-flap substrates, which form when a 5′ flap is generated with a 3′ single nucleotide flap occurring on the upstream strand [42–46]. Proteins in the FEN family have also been shown to have activity on a variety of other substrates, including nicked duplex DNA [29,32,43]. In all three kingdoms of life, organisms exist that encode multiple FENs [29,37,47]. In many bacteria, one of the FENs is a part of Pol I while the other exists as an independently encoded protein [22,29,35]. The FENs found as a domain of Pol I represent the most well-studied bacterial FENs to date, while the discrete FENs have received much less attention, making their contribution to DNA replication and genome instability unclear.

*B. subtilis* is one such organism known to encode a discrete FEN in addition to Pol I. This protein, YpcP, was initially identified as a protein homologous to the N-terminal domain of Pol I [22,48]. YpcP has been described as an exonuclease, and due to its similarity to the N-terminal domain of Pol I renamed ExoR [49]. Sequence homology and structural predictions of the Pol I N-terminal domain and YpcP show that both proteins belong to the wider FEN family [29]. For this reason, we will adopt the original nomenclature proposed for Pol I-independent, bacterial FENs [29] and herein refer to *B. subtilis* YpcP as FEN and its gene as *fenA*. Initial work characterizing the relationship between Pol I and FEN demonstrated that *B. subtilis polA* and *fenA* mutants are synthetically lethal [35,48] and that overexpression of *polA* or *fenA* was able to partially suppress the filamentous phenotype of an RNase H-deficient strain [35]. However, overexpression of *fenA* was unable to rescue a temperature sensitive strain lacking the N-terminal domain of Pol I [35]. Through these experiments, it was concluded that the two proteins have overlapping functions but are not redundant. Biochemical assays using purified FEN suggest that it has nucleolytic activity on a variety of substrates, with some preference shown for RNA-DNA hybrids [13]. Other evidence from protein-DNA binding assays suggests that FEN has the highest binding affinity for double-stranded DNA with a 5′ overhang, suggesting that FEN might prefer to act on DNA [49]. Together these data suggest that FEN may actively contribute to Okazaki fragment processing and DNA repair, although the relationship between Pol I and FEN remains unknown.

In this work, we examined the substrate preferences of Pol I and FEN using a variety of RNA-DNA hybrids and DNA-only substrates. We also examined the phenotype of *polA* and *fenA* during genotoxic stress and in RNase H deficient backgrounds to determine the contribution of Pol I and FEN 5′ nuclease activity to genome integrity. We found that FEN shows the most robust activity on RNA-DNA hybrid substrates modeling Okazaki fragment intermediates and that the strong *polA* phenotypes are rescued by simply expressing the C-terminal domain lacking 5′ nuclease activity. With our results, we conclude that discrete *B. subtilis* FEN functions as the major Okazaki fragment nuclease, with Pol I DNA polymerase activity

contributing to both DNA repair and Okazaki fragment resynthesis. Because discrete FENs are present in a wide-range of bacterial genomes, we suggest that Pol I-independent FENs provide the 5′ nuclease activity important for RNA removal during lagging strand replication for many bacteria. Further, our work suggests that lagging strand processing in many bacteria is more congruent with eukaryotes than previously thought.

## Results

### Loss of *fenA* confers sensitivity to DNA damage in the absence of ribonuclease HIII

Since *B. subtilis* encodes two proteins with hypothesized FEN activity, we began by investigating the importance of *fenA* for growth during conditions that result in DNA damage or dNTP depletion to test the model that FEN contributes to genome maintenance under stress conditions [49,50]. As shown in **Fig 1A**, a single deletion of *fenA* does not result in any detectable phenotype for cells grown on hydroxyurea (HU) or cells exposed to UV radiation (**S1 Fig**). It was previously shown that *fenA* disruption does not confer cold sensitivity nor sensitivity to mitomycin C (MMC) or methyl methanesulfonate (MMS) [13]. Given these results, it seems clear that *B. subtilis* FEN does not have an appreciable role during DNA repair *in vivo*. Another model is that FEN is involved in Okazaki fragment maturation. If true, we would expect a strain lacking both *rnhC* and *fenA* to confer a phenotype. Ribonuclease HIII (RNHIII), encoded by *rnhC*, is a protein known to participate in Okazaki fragment maturation and the resolution of RNA-DNA hybrids [12,13]. The absence of both proteins resulted in cells that were more sensitive to HU or UV than either single deletion (**Figs 1A and S1**), suggesting that FEN could be involved in the resolution of RNA-DNA hybrids on the lagging strand, and any defects due to the absence of FEN are exacerbated with the combined absence of RNHIII and presence of genotoxic stress. Notably, loss of *fenA* does not increase the sensitivity of a Δ*rnhB* strain (**Figs 1A and S1**), indicating that the sensitivity observed in the Δ*rnhC*, Δ*fenA* strain is due to a failure to repair RNA-DNA hybrids that occur from patches of ribonucleotides consistent with Okazaki fragments.

### FEN is reliant on residues conserved across bacterial FENs

FEN contains a conserved group of eight carboxylate resides that are predicted to be a part of the active site [29], which is modeled in **Fig 1B–1D**, with a multiple sequence alignment provided (**S2 Fig**). To test the importance of these amino acids, we designed a series of mutations to disrupt conserved residues and expressed each variant from the ectopic chromosomal *amyE* locus under the *Pspank* promoter, which allows induction of expression in the presence of isopropyl ß-D-1-thiogalactopyranoside (IPTG). As discussed above, the single deletion of *fenA* did not result in an obvious phenotype, so mutants were expressed in the Δ*rnhC*, Δ*fenA* background.

Overexpression of WT *fenA* rescued the double deletion phenotype (**Figs 1E and S3**). With the Δ*rnhC*, Δ*fenA* complementation assay established, four *fenA* mutants were created, two for each metal-binding site. The first mutant, *fenA*^E114Q,D116N^, was designed to abrogate binding of the Site 1 divalent cation without changing the overall shape of the active site. The binding at Site 1 was also altered in the second mutant, where the EADD motif (residues 114 to 117) was changed to AAAA, a construct designated as *fenA*^Site1^. In addition to loss of cation binding, we predict that this mutant would have alterations to the shape of the active site due to the hydrophobic residues, as well as changes to the flexible arch, since sequence similarity suggests that the D117 residue interacts with R82 to form a salt-bridge [51]. As shown in **Figs 1E and S3**,

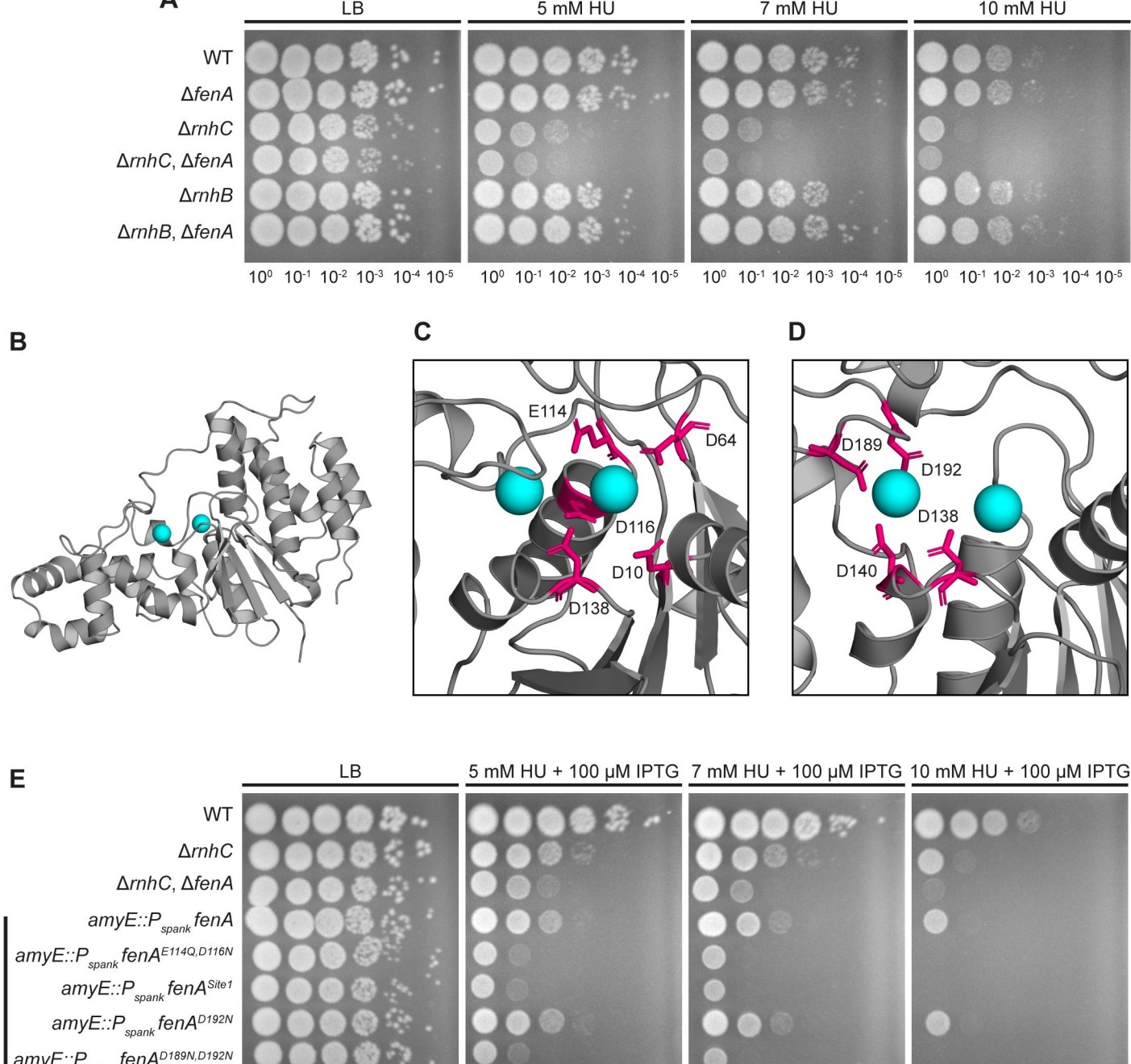

**Fig 1. FEN contributes to RNA-DNA hybrid repair *in vivo*.** (A) Spot titer assay to determine a phenotype of cells exposed to hydroxyurea (HU) in the absence of *fenA*. (B) Alphafold model of FEN, with (C) Site 1 conserved carboxylate residues and (D) Site 2 carboxylate residues shown in pink. Coordinated divalent metals are shown in blue and were added via alignment to *Mycobacterium smegmatis* FEN (PDB: 6C33). (E) Spot titer assay testing ectopically expressed mutants of *fenA* lacking conserved carboxylate residues for rescue of Δ*rnhC*, Δ*fenA* HU sensitivity.

neither mutant was able to rescue Δ*rnhC*, Δ*fenA* as well as WT *fenA*, suggesting that these residues are important for function of FEN *in vivo*. The third mutant, *fenA*^D192N^, has one residue from metal-binding Site 2 changed. Previous work suggested that this mutant was catalytically

inactive [13], however, overexpression in the double deletion strain rescues to nearly the same degree as the WT gene (**Figs 1E and S3**). Adding a second mutation, creating *fenA*$^{D189N,D192N}$, once again led to a failure to rescue the Δ*rnhC*, Δ*fenA* strain. Together this suggests that not every conserved residue is critical for activity, however, perturbations that significantly affect metal-binding render *fenA* inactive *in vivo*.

An observation from our experiments is that expression of *fenA*$^{E114Q,D116N}$ or *fenA*$^{D189N,D192N}$ in the Δ*rnhC*, Δ*fenA* strain results in cells that are more sensitive to HU than the parent strain or those expressing WT *fenA* (**Fig 1E**). To test if these mutations were dominant negative, we used the same system to express each mutant or the WT allele from an ectopic locus in WT or a Δ*rnhC* strain. There was no detectable phenotype in WT cells (**S4 Fig**), however, induced expression of either *fenA*$^{E114Q,D116N}$ or *fenA*$^{D189N,D192N}$ in the Δ*rnhC* background resulted in cells that were demonstrably more sensitive to HU than the parent strain (**S5 Fig**). Even when *fenA*$^{E114Q,D116N}$ or *fenA*$^{D189N,D192N}$ were induced on LB plates with no stressor, the colonies were smaller than those of the parent strain (**S5 Fig**). We speculate that the metal-binding variants are still able to bind substrates *in vivo* but are unable to bind one of the metal ions and therefore lack the catalytic activity required for turnover. We hypothesize that in WT cells, the mutants are out-competed by the overlapping functions of Pol I, FEN, and RNHIII, and cause no phenotype. Loss of RNHIII disrupts this balance, which is then exacerbated by exposure to HU [52], ultimately leading to a phenotype during mutant overexpression.

## FEN is more active on canonical double-flap structures than Pol I

The *fenA* phenotype supports a contribution to Okazaki fragment processing as opposed to DNA repair. Given this, we chose to assay FEN and Pol I *in vitro* to compare their nuclease activity on a series of substrates to identify the substrates preferred by each enzyme. To do so, we expressed and purified each protein with a His$_6$-SUMO tag, which we cleaved prior to use (detailed in Methods). This allowed purification of full-length proteins without any additional residues (**S6 Fig**). Using fluorescently labeled oligonucleotides (**S1 Table**), we generated a series of biologically relevant substrates. Each substrate had a DNA-only version, as well as a chimera / RNA-DNA hybrid version, where the first 12 bases at the 5′ end of the labeled strand are equivalent ribonucleoside monophosphates (rNMPs). In **Figs 2–7**, the assayed substrates are depicted at the top of the gels, with ribonucleotides indicated by zig-zag lines. As the activity of FENs is dependent on the divalent cations at the protein's active site, we used a buffer containing physiologically relevant concentrations of Mg$^{2+}$ and Mn$^{2+}$ as described previously [12], which were based on studies of free magnesium [53] and estimated values necessary for magnesium and manganese transporter activation in *B. subtilis* [54,55].

The first substrates we tested were variations of a 5′ flap, as this structure can be generated *in vivo* during strand-displacement synthesis by Pol I [56]. This structure is often associated with Okazaki-fragment maturation [27,28], nucleotide excision repair (NER) or long patch base excision repair (BER) [57]. The first substrate tested was a nicked double-flap, composed of a single nucleotide 3′ flap directly abutting a downstream 5′ flap (**Fig 2**). This is the canonical structure associated with FEN activity, as some FENs have been shown to contain a 3′ binding pocket, capable of binding a single displaced nucleotide from the upstream DNA strand [58–61]. Utilization of this pocket stimulates protein activity, increasing substrate turnover [58,60]. As demonstrated by **Fig 2A**, FEN is capable of cleaving both the RNA-DNA hybrid and DNA-only versions of the substrate, with a preference for the hybrid as quantified below the gel. The FEN$^{D192N}$ mutant was previously suggested to be catalytically inactive because it showed no activity when assayed at a lower concentration [13]. However, at a higher concentration, FEN$^{D192N}$ generated similar products to those of the WT FEN, although it had reduced

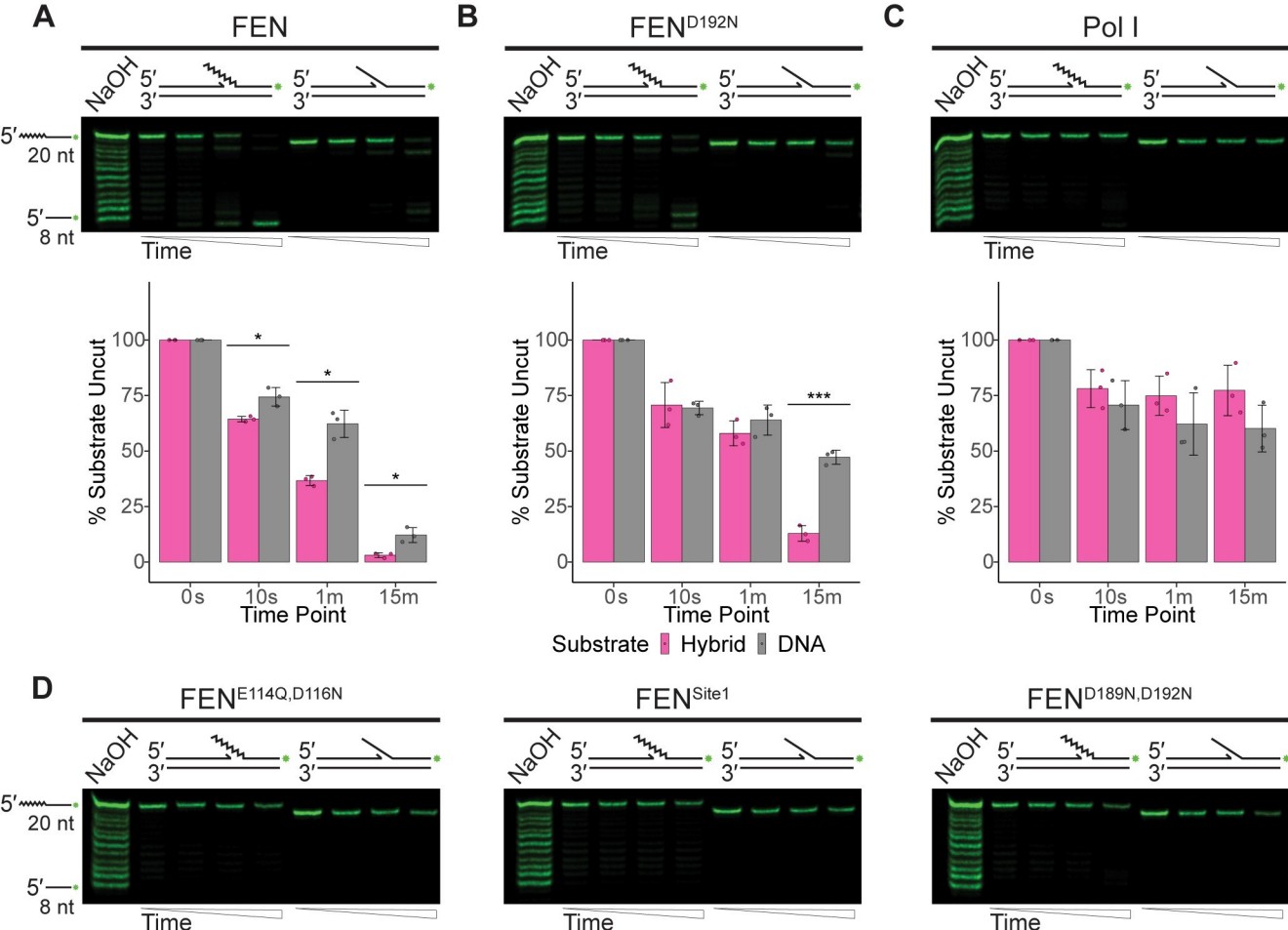

**Fig 2. FEN is more active on canonical double-flap substrates than Pol I.** (A) Products of nuclease activity of FEN, (B) FEN[D192N], or (C) Pol I on nicked, 5′ flaps with an abutting single nucleotide 3′ flap resolved using denaturing urea-PAGE. The mean for percentage of intact substrate across three replicates is graphically shown beneath the respective gel with error representing the standard deviation. Asterisks indicate significance as follows: * $p < 0.05$, ** $p < 0.01$, or *** $p < 0.001$. (D) Representative gels of FEN[E114Q,D116N], FEN[Site1], or FEN[D189N,D192N] activity on the described substrate from at least three replicates. For all assays, substrates consist of oligonucleotides oFCL11 and oJR368 with oJR339 (RNA-DNA hybrid; RNA indicated by zigzags) or oJR348 (DNA-only). Time points for all assays are 0 s, 10 s, 1 min, and 15 min with ladders generated by alkaline hydrolysis.

activity on both substrates (**Fig 2B**). This is consistent with the *in vivo* results shown in **Figs 1E and S3**. Surprisingly, Pol I had minimal activity on either version of the canonical substrate (**Fig 2C**). The other three FEN mutants surveyed in **Fig 1** were also assayed on the nicked double-flap substrate and had no significant activity on either variant (**Fig 2D**). We conclude that FEN is much more active on the canonical substrate compared with Pol I.

## FEN and Pol I have similar activity on gapped double-flap structures

The next substrate we investigated was like the previously described canonical structure, although the upstream and downstream pieces were separated by a four-nucleotide gap. WT FEN had strong activity on both substrate types, although it preferred the RNA-DNA hybrid (**Fig 3A**). The FEN[D192N] mutant behaved similarly (**Fig 3B**), and the cleavage pattern of the hybrid suggests that FEN is less specific with the incision point of this substrate and may engage more in exonuclease activity. In a stark contrast to the nicked double-flap, Pol I had high activity on both the RNA-DNA hybrid and DNA-only substrates when the flaps were

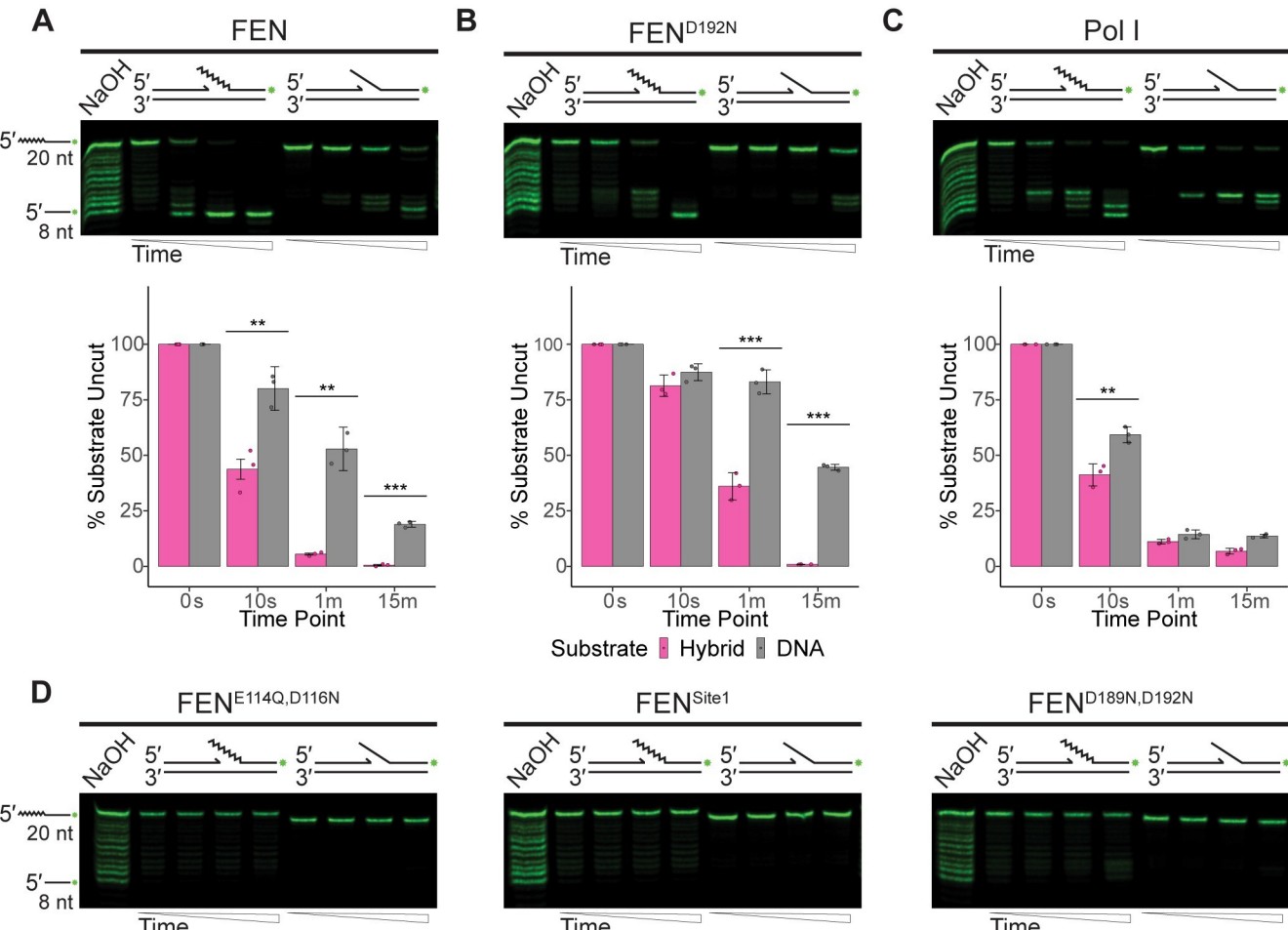

**Fig 3. FEN and Pol I are active on gapped double-flap substrates.** (A) Nuclease assay results of FEN, (B) FEN<sup>D192N</sup>, (C) or Pol I on gapped double-flap substrates separated by urea-PAGE. The mean percentage of uncut substrate is quantified under the respective gel, based on three replicates with error bars representing the standard deviation. Significance is indicated as follows: * $p<0.05$, ** $p<0.01$, or *** $p<0.001$. (D) Gels of FEN<sup>E114Q,D116N</sup>, FEN<sup>Site1</sup>, or FEN<sup>D189N,D192N</sup> activity on the gapped double-flap substrate. Gels are representative of at least three replicates Substrates were made by annealing oFCL10 and oJR368 with oJR339 (RNA-DNA hybrid; RNA indicated by zigzags) or oJR348 (DNA-only). Time points for all assays are 0 s, 10 s, 1 min, and 15 min with ladders obtained by incubating the hybrid structure with sodium hydroxide.

separated by a gap (**Fig 3C**). Even with this substrate there is a clear difference in the distribution of products produced by FEN and Pol I. FEN shows complete cleavage of the flap at earlier timepoints as compared with Pol I which engages in exonucleolytic degradation of the flap. The FEN<sup>E114Q,D116N</sup> and FEN<sup>Site1</sup> mutants did not have significant activity on the substrates, however, the FEN<sup>D189N,D192N</sup> mutant did have some weak activity on the hybrid substrate (**Fig 3D**). FEN<sup>D189N,D192N</sup> activity was minimal, even compared to the FEN<sup>D192N</sup> mutant, but suggests that our abrogation of the second metal binding site did not result in a complete loss of catalytic activity on the gapped double flap substrate.

## FEN is more active on 5′ flap substrates than Pol I

We next tested a 5′ flap overhang with a four-nucleotide gap between the upstream piece and the flap junction (**Fig 4**). Unlike the prior substrates, this substrate lacks the 3′ flap. As shown in **Fig 4A**, FEN was able to cleave both versions of this substrate well, rapidly generating a low molecular weight product for both, although it had significantly more activity on the hybrid

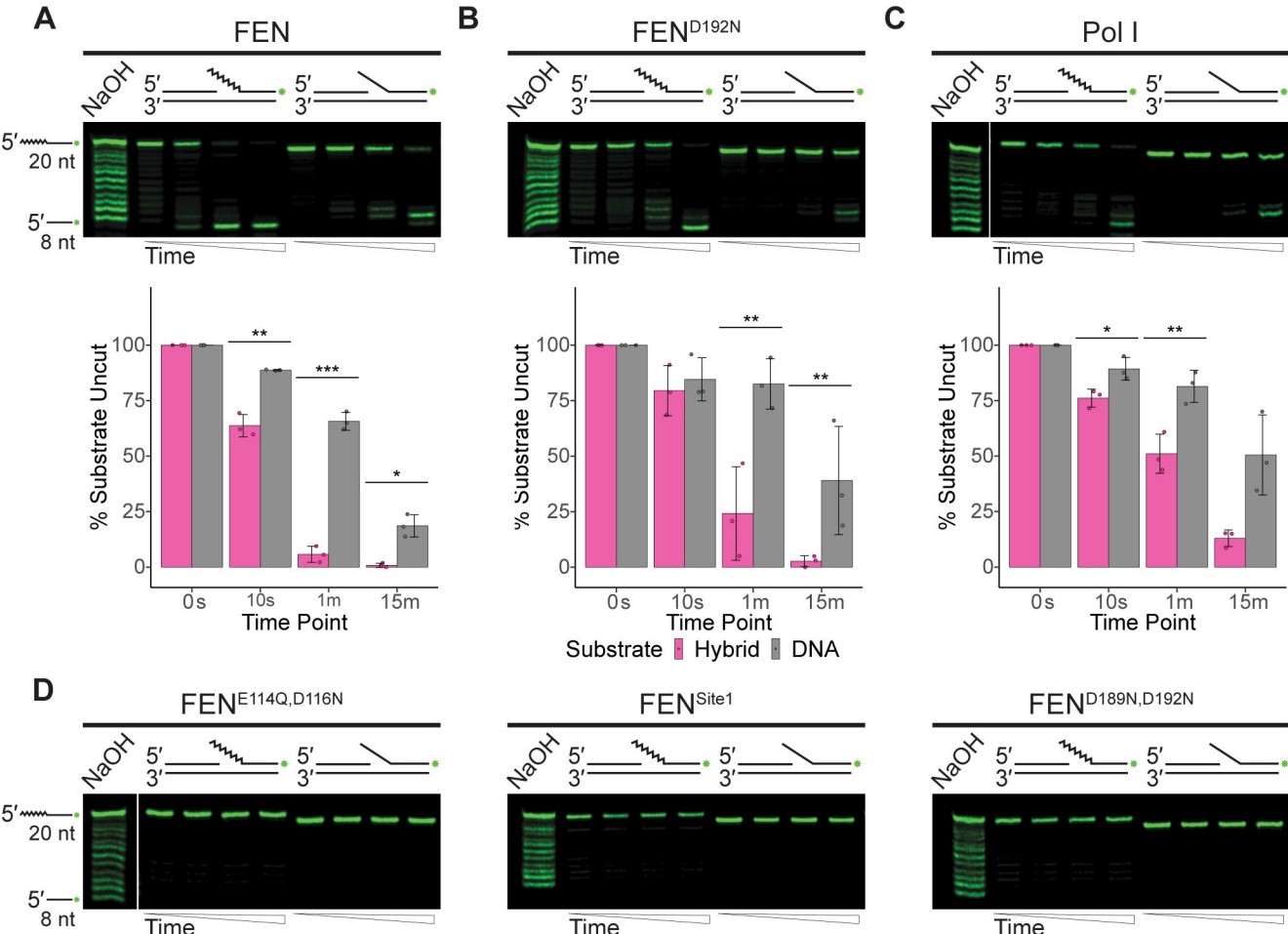

**Fig 4. FEN is active on flap overhang structures.** (A) Endonuclease activity of FEN, (B) FEN[D192N], or (C) Pol I on 5′ flap overhang structures visualized using denaturing urea-PAGE. The mean percentage of substrate remaining uncut by each protein across three replicates is quantified with standard deviation shown beneath a representative gel; asterisks indicate significance: * p<0.05, ** p<0.01, or *** p<0.001. (D) Assays of activity on 5′ flap structures by the catalytic mutants FEN[E114Q,D116N], FEN[Site1], or FEN[D189N,D192N]. For all assays, flap substrates consist of oligonucleotides oJR366 and oJR368 with oJR339 (RNA-DNA hybrid; RNA indicated by zigzags) or oJR348 (DNA-only). Time points for all assays are 0 s, 10 s, 1 min, and 15 min with ladders generated by alkaline hydrolysis.

flap structure. The FEN[D192N] mutant was also active on both substrates (**Fig 4B**), although they produced less product than the wild-type protein. Like FEN, Pol I preferred the hybrid flap substrate and generated a product consistent with flap endonuclease activity, however overall it had much lower catalytic activity than FEN (**Fig 4C**). The three remaining mutants, FEN[E114Q,D116N], FEN[Site1], and FEN[D189N,D192N] show no noticeable activity on the substrate, consistent with their failure to rescue ΔrnhC, ΔfenA strains (**Fig 4D**). The activity of Pol I on the DNA substrate is more indicative of a role in DNA repair while the activity of FEN, cleaving the RNA-DNA hybrid to completion, is more indicative of a role for FEN in Okazaki fragment maturation.

Previous work has shown that some FENs have increased activity when potassium ions are bound to the protein [40,41]. We also assayed for nuclease activity on this simple flap structure in the presence of potassium. As shown in **S7 Fig**, neither FEN nor Pol I was appreciably stimulated by the addition of potassium to the reaction mixture.

## FEN is more active on nicked substrates than Pol I

After testing the series of flapped intermediates, we investigated nuclease activity on a nicked substrate which would occur prior to flap formation (**Fig 5**). Nicks are a common type of DNA damage, where a missing phosphodiester bond occurs in otherwise continuous DNA. Nicks can result from many DNA repair pathways, including NER, BER, and mismatch repair [62]. Nicks can also be created following RNase H activity [63] or during Okazaki fragment maturation [64]. FEN showed high activity on the RNA-DNA hybrid nick substrate (**Fig 5A**), fully removing the rNMPs. While FEN was also fully active on the DNA-only substrate, it appears to use primarily 5′-3′ exonuclease activity, resulting in a sequential removal of nucleotides from the 5′ end rather than the single band associated with flap endonuclease activity. Regardless of the specific type of incision used, FEN has an overall preference for the hybrid structure. The FEN[D192N] mutant (**Fig 5B**) shows both flap endonuclease and 5′ exonuclease activities on the hybrid structure, but not at the same rate observed for the WT enzyme. Despite the differences in activity, FEN and the FEN[D192N] mutant generate similar products for each substrate

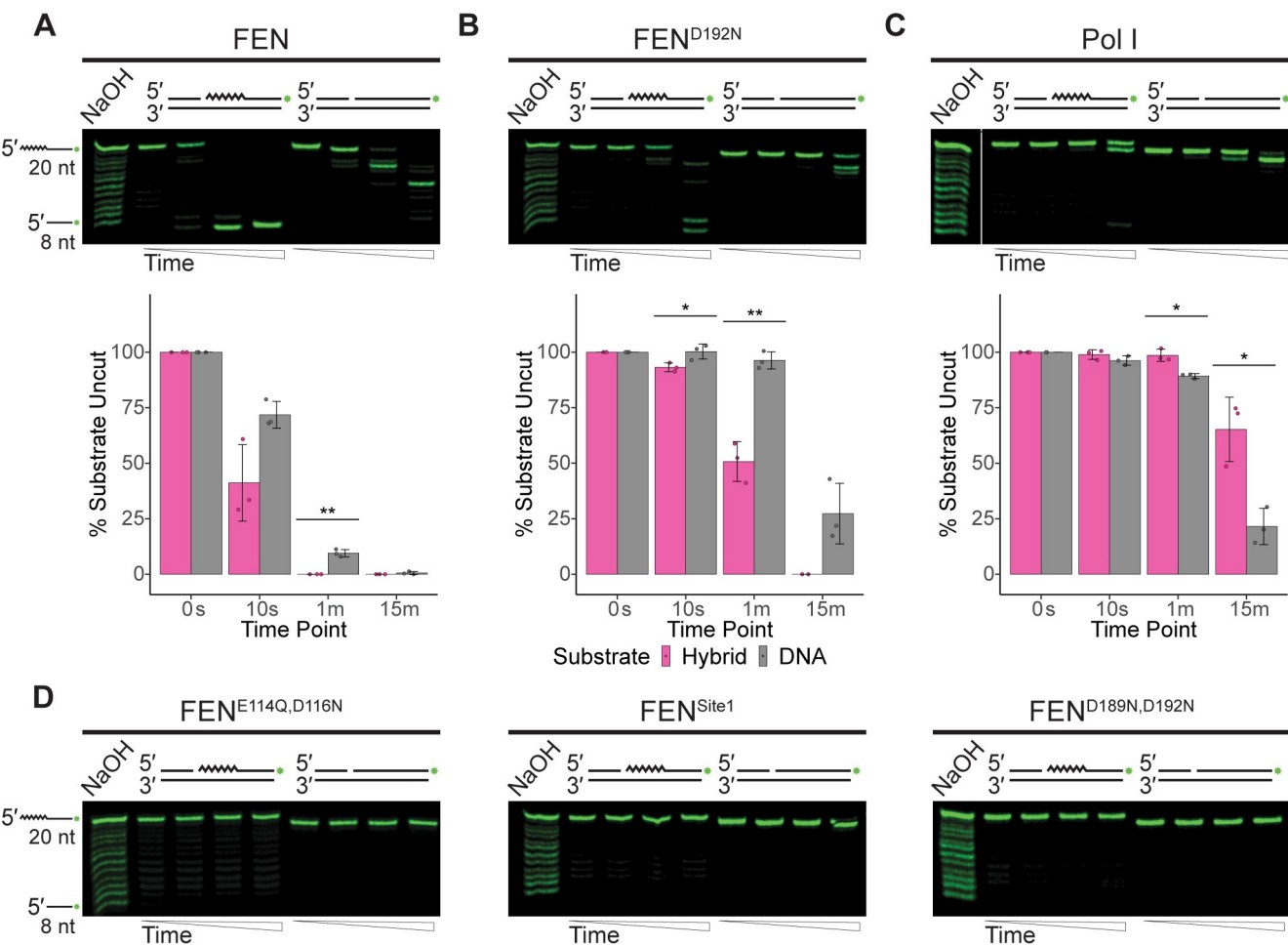

**Fig 5. FEN can remove RNA from Okazaki fragment-like nicks.** (A) Visualization of nuclease activity of FEN, (B) FEN[D192N], or (C) Pol I on nicked duplex substrates. Three replicates of each assay are quantified beneath a representative urea-PAGE gel with standard deviation shown. Asterisks are used to indicate significance: * p<0.05, ** p<0.01, or *** p<0.001. (D) Results of incubation of nicked duplex substrates with the FEN mutants FEN[E114Q,D116N], FEN[Site1], or FEN[D189N,D192N]. Nicked duplex substrates consist of oJR338 and oJR340 with oJR339 (RNA-DNA hybrid; RNA indicated by zigzags) or oJR348 (DNA). Ladders were generated by treating hybrid structure with sodium hydroxide. Assay time points for all reactions are 0 s, 10 s, 1 min, and 15 min.

by the 15-minute time point. Conversely, Pol I had much less activity on either variant of the nicked substrate than FEN (**Fig 5C**). Pol I engaged minimally in 5′ exonuclease activity on each nick, removing single nucleotides from each substrate. Interestingly, Pol I had significantly more activity on the DNA-only substrate than the hybrid, demonstrating a preference opposite to that shown by FEN. As before, the remaining FEN mutants appear catalytically inactive and do not have detectable activity on the nicked substrate (**Fig 5D**).

## FEN is more active on 3′ overhang structures than Pol I

The final Okazaki fragment-like substrate we tested was a 3′ overhang (**Fig 6**). This substrate is primarily associated with Okazaki fragment maturation [15] but can be formed during other DNA processes [65]. FEN was highly active on the hybrid version of this substrate, which most closely mimics an Okazaki fragment, cleaving all rNMPs from more than 80% of the substrate within 10 seconds (**Fig 6A**). FEN was also active on the DNA-only substrate, although the cleavage pattern suggests that this is primarily 5′ exonuclease activity. As with the other

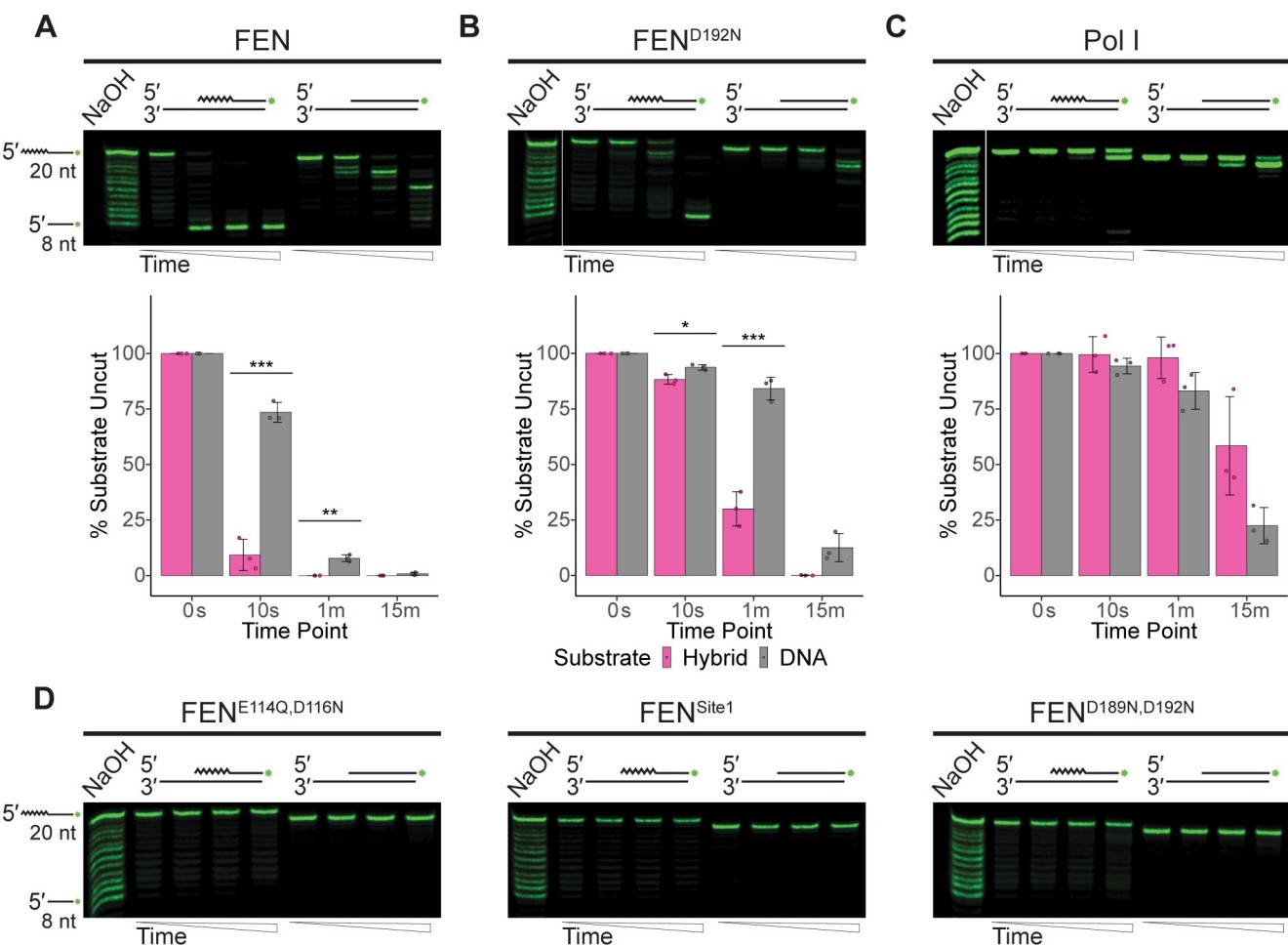

**Fig 6. FEN is more active on 3′ overhangs than Pol I.** (A) Nuclease activity of FEN, (B) FEN[D192N], or (C) Pol I on 3′ overhang structures shown using 20% urea-PAGE. The mean percent of substrate left uncut by the protein, based on three replicates, is shown beneath the respective gel. Significance is indicated by asterisks: * p<0.05, ** p<0.01, or *** p<0.001. Standard deviation across replicates is indicated by the black bars. (D) Failure of FEN mutants FEN[E1414Q, D116N], FEN[Site1], and FEN[D189N,D192N] to cleave 3′ overhang substrates. Representative gels of at least three replicates are shown. Substrates assayed were generated by annealing oJR340 with oJR339 (RNA-DNA hybrid; RNA indicated by zigzags) or oJR348 (DNA). The triangle indicates time point progression of 0 s, 10 s, 1 min, and 15 min; the ladder was produced via alkaline hydrolysis of the hybrid substrate.

substrates, the FEN[D192N] mutant had reduced activity on both the hybrid and DNA-only substrates, producing products consistent with those of WT FEN (**Fig 6B**). Pol I had minimal exonuclease activity on either substrate, engaging primarily in 5′ exonuclease activity (**Fig 6C**). Despite the high activity of WT FEN on the 3′ overhang, the remaining FEN mutants were not demonstrably active, further highlighting the essentiality of these residues to FEN function (**Fig 6D**). These substrates: 5′ flaps (with or without a 3′ flap), nicked duplex DNA, and 3′ overhangs, represent the most common structures associated with Okazaki fragment maturation. In all cases, FEN is more active than Pol I on the RNA-DNA hybrid versions of the substrates tested. Based on these data, we suggest that FEN contributes significantly to the resolution of Okazaki fragments *in vivo*.

## FEN is more active on blunt substrates than Pol I

In addition to Okazaki fragment-like substrates, we assayed the proteins for activity on less canonical substrates. The first of these (**Fig 7**) mimicked blunt DNA, which can occur *in vivo* after a nick is converted to a double-stranded break when encountered by a replication fork [66]. As **Fig 7A** shows, FEN had high activity on both the hybrid and DNA-only substrates, with a significant preference for the hybrid. Furthermore, FEN demonstrates different activities on the two variants of the substrate: on the hybrid it showed endonuclease activity and cleaved all the rNMPs while on the DNA-only version it cleaved exonucleolytically. The catalytically reduced FEN[D192N] mutant produced similar results (**Fig 7B**), although the enzymatic efficiency was reduced. Pol I had no remarkable activity on either substrate (**Fig 7C**). The FEN mutants were not active on the blunt substrates (**Fig 7D**).

## Pol I has more activity on 5′ overhangs than FEN

We also tested activity on a 5′ overhang structures (**S8 Fig**), which make up a portion of double-stranded breaks in the cell [66]. While not as striking as with the previous substrates, FEN showed exonuclease activity on both the hybrid and the DNA-only substrate (**S8A Fig**). This activity was not detected with the FEN[D192N] mutant (**S8B Fig**), suggesting that the 5′ overhang is not a preferred substrate of FEN. The remaining FEN mutants also had no discernable activity on the substrates (**S8D Fig**). Unlike FEN, Pol I was able to perform endonucleolytic cleavage on both the hybrid structure and the DNA-only 5′ overhang (**S8C Fig**). Pol I had overall lower activity on the DNA-only substrate than the hybrid, however, a distinct band indicative of the removal of multiple nucleotides was present for both. The differences between the activities of FEN and Pol I on the 5′ overhang substrate are striking. Interestingly, FEN from bacteriophage T5 was shown to have activity on 5′ overhangs as seen here with Pol I, while FEN from *Staphylococcus aureus* behaved similarly to *B. subtilis* FEN [29]. This leads us to conclude that, while FEN and Pol I belong to the same family, these proteins have distinct roles in maintaining genome integrity.

## FEN has more activity on single-stranded substrates than Pol I

The last construct that we investigated is a single-stranded substrate with no double-stranded regions (**S9 Fig**). It has been established that *Ec*Pol I 5′ to 3′ exonuclease activity requires duplex DNA [67], however this is not the case for all FEN proteins [29]. FEN had exonuclease activity on both versions of this substrate, but there was no detectable band indicative of full removal of the rNMPs (**S9A Fig**). The FEN[D192N] mutant behaved similarly to WT FEN, but with reduced overall activity (**S9B Fig**). Pol I had negligible activity on either variant and did not create a distinct product, in agreement with prior work showing that neither Pol I from *E. coli* nor *Thermus aquaticus* is active on ssDNA [67,68]. While the three catalytically inactive

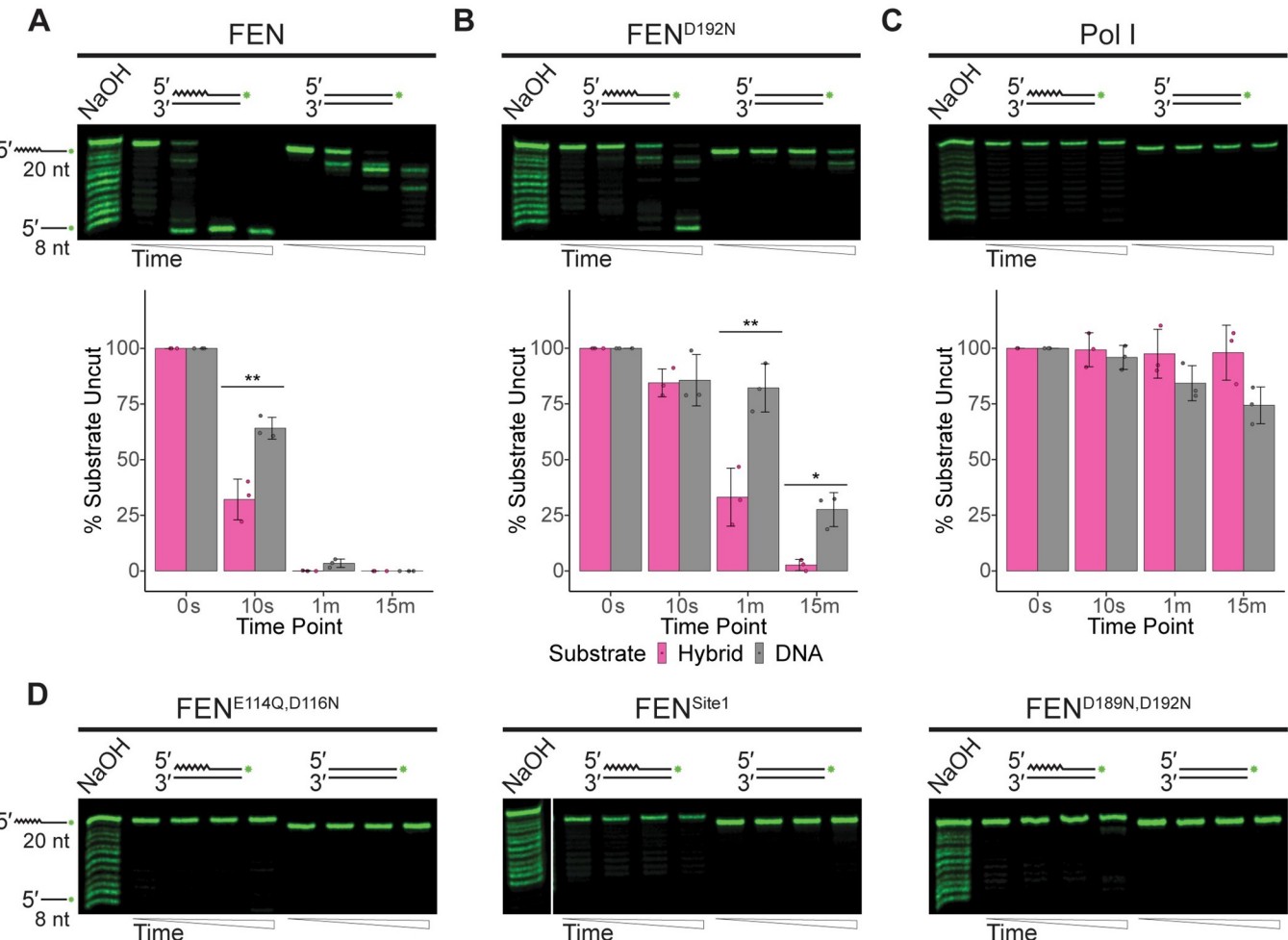

**Fig 7. FEN has strong activity on blunt duplex substrates.** (A) Denaturing urea-PAGE gels showing assays of FEN, (B) FEN^D192N, or (C) Pol I activity on blunt duplex structures. The mean percent uncut substrate remaining from three replicates is quantified under the appropriate gel. Black bars represent standard deviation and asterisks symbolize significance: * p<0.05, ** p<0.01, or *** p<0.001. (D) Assays of FEN mutants: FEN^E1414Q,D116N, FEN^Site1, and FEN^D189N,D192N on the blunt substrates. Representative gels of at least three replicates are shown. Substrates were composed of oligonucleotides oJR365 and either oJR339 (RNA-DNA hybrid; RNA indicated by zigzags) or oJR348 (DNA only), with a ladder generated by treating the hybrid version with sodium hydroxide. Time points for all reactions are 0 s, 10 s, 1 min, and 15 min.

FEN mutants initially appear to have some activity on this substrate (**S9D–S9E Fig**), assays lacking protein suggest that minor loss of signal is attributable to background degradation of the RNA substrate (**S9G Fig**) as it is more prone to spontaneous hydrolysis when single-stranded. We conclude that FEN is active on ssDNA and ssRNA/DNA hybrids.

## Pol I nuclease activity is not stimulated with extension

Given the striking differences in the nuclease activities of FEN and Pol I, we asked whether these differences could be attributed to the presence of the Klenow fragment on Pol I. While FEN is composed of a single domain, Pol I also has the Klenow fragment, which is approximately twice as large as the Pol I FEN-domain. In the established model for Okazaki fragment maturation, there must be coordination between the FEN-domain and the Klenow fragment [21,64,69]. To account for this, we tested Pol I activity on a dual-labeled substrate mimicking an Okazaki fragment. As **Fig 8** (left) shows, in the absence of dNTPs, Pol I was unable to extend

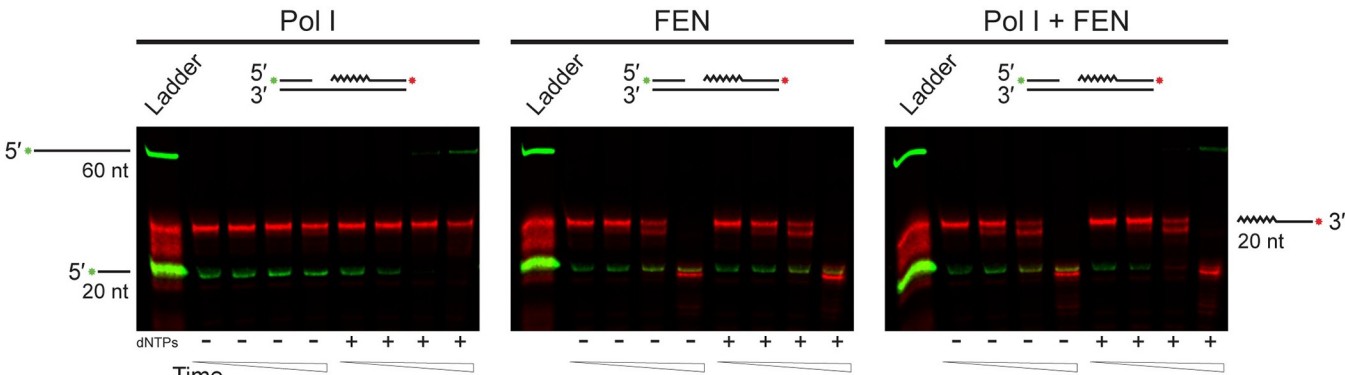

**Fig 8. Pol I has lower nuclease activity than FEN even with concurrent DNA synthesis.** Denaturing urea-PAGE of products generated by Pol I or FEN activity on a substrate resembling an Okazaki fragment, where upstream and downstream fragments are separated by ten bases. Extension by synthesis from the upstream, 3′ hydroxyl group is tracked by a 5′ fluorescent labeled substrate (green, oFCL8) while nuclease activity on the downstream Okazaki-fragment is tracked with the red colored, 3′ labeled oligonucleotide (oJR367). Labeled oligonucleotides were annealed to oFCL6. Each protein was tested with or without 50 μM dNTPs and time points of 0 s, 10 s, 1 min, and 15 min. Ladder was generated by adding oFCL11 annealed to oFCL6 to sodium hydroxide treated substrate. Images are representative of at least three replicates.

the free 3′ end of the 5′-labeled substrate and did not noticeably cleave the 3′-labeled substrate. In the presence of dNTPs, Pol I was able to extend the 5′-labeled substrate and cleave minimal downstream nucleotides, while most of the substrate remained intact. We assayed FEN under the same conditions **Fig 8** (middle) and found that it cleaved the primer-like portion of the 3′-labeled substrate regardless of whether dNTPs were included in the reaction buffer. Combining both proteins at equimolar concentrations **Fig 8** (right) resulted in extension of the 5′-labeled substrate and cleavage of the 3′-labled substrate. Thus, even under conditions that promote engagement of both activities of Pol I, FEN had greater catalytic activity on the substrate, which is consistent with our biochemical characterization of FEN. Together, our biochemical work suggests that FEN is the major contributor to the repair of Okazaki fragments in *B. subtilis*.

### ΔpolA strain sensitivity is due to loss of the Klenow fragment

Unlike the Δ*fenA* strain described earlier, a single deletion of *polA* results in sensitivity to DNA damage from MMC and MMS [13,23,35]. It has been shown in *Streptococcus pneumoniae* and *Haemophilus influenzae* that 5′ to 3′ FEN of Pol I is essential, although these two bacteria lack *fenA* or any other protein with a FEN domain apart from Pol I [29,70,71]. Given our biochemical results, we hypothesized that the Δ*polA* phenotype is due to the loss of the Klenow fragment rather than loss of the FEN-domain. Since prior work has shown that Pol I activity can be reconstituted even with the fragments physically separated [17], we constructed IPTG-inducible strains to express Pol I (*polA*), the Pol I FEN-domain (*polA_FEN*), or the Pol I Klenow fragment (*polA_Klenow*) in the Δ*polA* background. As shown in **Fig 9**, overexpression of either full-length Pol I or the Klenow fragment was sufficient to rescue the *polA* deletion strain. Overexpressing the Pol I FEN domain did not reduce MMC sensitivity, indicating that the strong phenotypes associated with Δ*polA* strains exposed to DNA damage is due to loss of polymerase activity rather than loss of nuclease activity. This result reaffirms that the FEN domain of Pol I is not critical for normal cell growth, and that FEN is sufficient even when cells are exposed to DNA damage.

### Pol I cannot directly substitute for FEN

We have established that FEN has more nuclease activity than Pol I on biologically relevant substrates assayed *in vitro*. We also considered the implications that this might have in the cell.

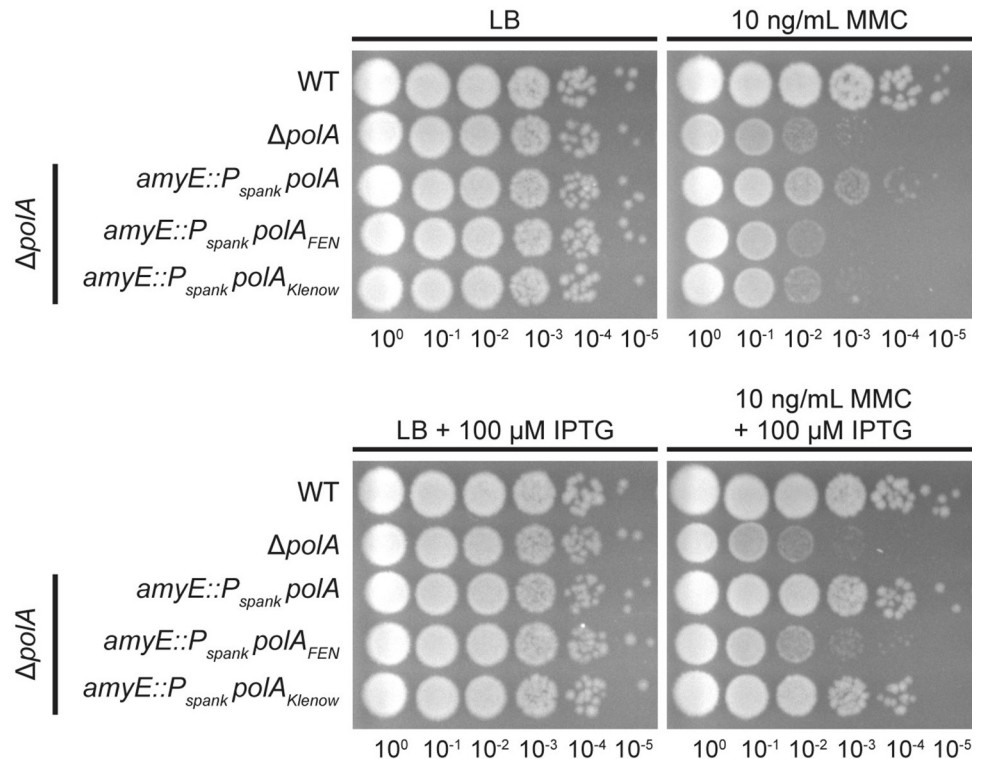

**Fig 9. Sensitivity of *polA* deletion strains is due to loss of the Klenow fragment.** Rescue of *polA* deletion strain sensitivity to mitomycin C (MMC) by ectopic, full-length Pol I (*polA*) or Pol I polymerase domains (*polA*$_{Klenow}$ and *polA*$_{FEN}$). Plates were incubated at 30˚C.

As **Fig 1E** demonstrated, overexpression of FEN or the catalytically compromised FEN$^{D192N}$ mutant was able to rescue the Δ*rnhC*, Δ*fenA* strain. This led us to question if the biochemical difference we show between FEN and Pol I was realized in cells. We created Δ*rnhC*, Δ*fenA* strains expressing IPTG-inducible *fenA*, *fenA*$^{D192N}$, *polA*, *polA*$_{FEN}$, or *E. coli xni* and assayed for rescue of HU sensitivity. As before, overexpression of either *fenA* or *fenA*$^{D192N}$ resulted in rescue (**Fig 10**). Neither the FEN domain of Pol I nor full-length Pol I were able to rescue the strain as well as *fenA* or *fenA*$^{D192N}$, indicating that FEN's function in cells cannot be fully compensated for by Pol I alone. Expression of *xni*, which encodes the *E. coli* FEN paralog ExoIX, also failed to rescue the strain and instead showed a small dominant negative phenotype, likely caused by binding SSB in *B. subtilis* or causing an inactive complex on substrates *in vivo*. This protein has been shown to have some catalytic activity [41] although it is missing three of the carboxylate residues that typify bacterial FENs (**S2 Fig**); as such, it may have reduced activity compared to FENs that contain all eight residues [29]. We conclude that the high catalytic activity of FEN cannot be directly substituted for by Pol I, especially in the absence of RNase HIII.

## Discussion

FEN, previously referred to as YpcP or ExoR, is a member of the large FEN family of proteins that are found across all domains of life [45,72]. These proteins play an essential role in genome maintenance, particularly during Okazaki fragment maturation as FEN activity is required for viability [73]. For bacteria including *Streptococcus pneumonia* and *Haemophilus influenzae*, the Pol I FEN domain is essential although these organisms lack a second FEN

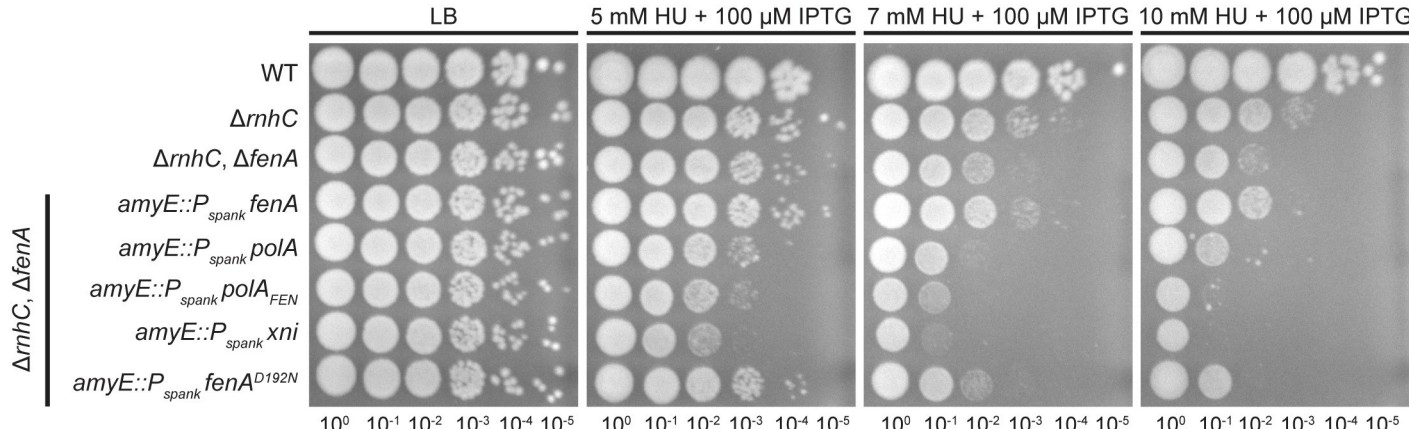

**Fig 10. Expression of *polA* does not rescue deletions of *fenA*.** Ectopic expression of different proteins with FEN activity in a Δ*rnhC*, Δ*fenA* strain sensitive to hydroxyurea (HU). *fenA* and *polA* encode the full-length proteins from *B. subtilis*, *fenA*$^{D192N}$ and *polA*$_{Klenow}$ express derivatives of the native protein, and *xni* encodes the *E. coli* homolog of FEN, ExoIX, which lacks three of the carboxylate residues associated with bacterial FENs.

[29,70,71]. In the case of *B. subtilis*, an organism that encodes two proteins with FEN activity, at least one must be present for viability [35]. Our biochemical analysis reveals surprising differences between the nuclease activities of FEN and Pol I. We show that FEN has more nuclease activity than Pol I on substrates that resemble Okazaki fragments. This is the case even for canonical structures associated with Pol I activity, nicked double-flaps, 5′ flaps, nicks, and 3′ overhangs, with FEN showing more activity on RNA. In fact, across all tested structures except the 5′ overhang, FEN showed a strong preference for RNA-DNA hybrids over dsDNA further suggesting that FEN is the dominant nuclease during Okazaki fragment processing in *B. subtilis*. We conclude that FEN's major biological function is the removal of RNA primers during lagging strand synthesis. There is evidence suggesting that FEN contributes to DNA repair, particularly during UV damage [49,50]. Given the poor activity of FEN on DNA substrates and the lack of a DNA damage phenotype for Δ*fenA*, we conclude that FEN does not provide a significant contribution to the repair of UV-induced DNA damage.

An important consideration regarding the activities of FEN and Pol I is that the latter is physically constrained by the Klenow fragment. The two major activities of Pol I are present in separate domains, however they are joined by a flexible linker. Biophysical modeling suggests that the Klenow fragment has lower affinity for the flap structure, which allows it to be replaced by the FEN domain [21,69,74,75]. To account for potential interference by the Klenow fragment, we also assayed for nuclease activity under conditions where extension occurs from an upstream 3′ hydroxyl. While extension by Pol I occurred in the presence of dNTPs, the nuclease activity of Pol I remained much lower than FEN under the same conditions. The relatively low nuclease activity of Pol I that we observe is consistent with other studies, as purifications of Pol I from *B. subtilis* were noted to have lower levels of nuclease activity than purifications from *E. coli* [26,76]. It is important to note that these early results were possibly influenced by proteolysis during protein purification [17,27,77]; to avoid this, the proteins used in this study were confirmed to be full length using SDS-PAGE (**S6 Fig**). Additionally, our group has previously shown that cleavage of an Okazaki fragment-like substrate by Pol I is minimal, and that repair of that substrate is increased when RNase HIII is included in the reaction [13]. Together, our biochemical data show that FEN is the major nuclease during Okazaki fragment maturation in *B. subtilis*, a role previously attributed to DNA polymerase I.

Despite FENs apparent role in Okazaki fragment maturation, Δ*fenA* cells do not demonstrate a detectable phenotype, though a Δ*rnhC* Δ*fenA* strain is more sensitive to genotoxic stress than either of the single deletions. With these results, we suggest that FEN is actively involved in the resolution of RNA-DNA hybrids, although some of this activity can be compensated for by other repair pathways [13]. Unlike the Δ*fenA* strain, Δ*polA* strains are sensitized to DNA damage, including damage caused by MMC and UV [13,23,35]. Given our biochemical results, we suspect that this is due to loss of the polymerization activities of Pol I rather than loss of the nuclease. Indeed, overexpression of the Klenow fragment rescued the Δ*polA* strain as well as full-length *polA*. Similarly, MMS sensitivity in *E. coli* was attributed to mutations affecting the polymerase activity of Pol I [25,78]. Given that Pol I is known to be broadly involved in the repair of DNA damage [62], it is likely that loss of the Klenow fragment and its strand-displacement synthesis activity leads to an accumulation of unrepaired DNA, resulting in genotoxic stress.

It has been accepted that Pol I is the major polymerase involved with Okazaki fragment maturation in bacteria [15,18], due to its dual function as polymerase and nuclease [15]. We show that Δ*rnhC*, Δ*fenA* strains are best rescued by WT FEN or a mutant with reduced catalytic activity (FEN$^{D192N}$), and that overexpression of Pol I or Pol I$_{FEN}$ fails to rescue the strain. Our work complements the results of Fukushima *et al.* [35], and further advances the idea that, while FEN and Pol I can degrade similar substrates, the proteins have different contributions and substrate preferences in the cell. We suggest that the removal of RNA primers during Okazaki fragment maturation in *B. subtilis* is carried out primarily by FEN while the upstream Okazaki fragment is extended by Pol I, as modeled in **Fig 11**. In the absence of FEN, the combined activities of Pol I and RNHIII could operate as a secondary pathway for maturation of the fragments yielding WT growth. We show that Pol I has levels of biochemical activity very similar to FEN$^{D192N}$, yet *fen*$^{D192N}$ complements the Δ*rnhC*, Δ*fenA* phenotype better than *polA*.

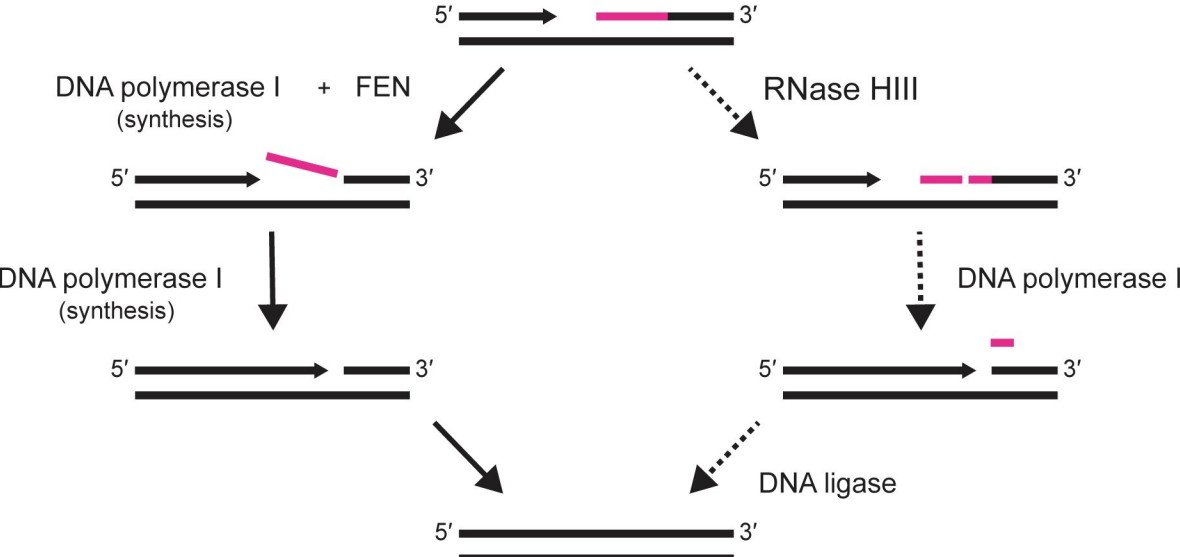

**Fig 11. A model of the proposed role of FEN in maturation of Okazaki fragments.** During lagging strand synthesis, maturation of Okazaki fragments can occur via two pathways. In the primary pathway (left, solid arrows), FEN removes RNA primers (pink) from Okazaki fragments while Pol I synthesizes from the upstream 3′ hydroxyl. DNA synthesis continues until Pol I loses affinity for the substrate and is outcompeted by DNA ligase, which seals the remaining nick. In the alternate pathway (right, dashed arrows), RNase HIII internally cleaves the RNA primer, shortening the RNA-DNA hybrid to facilitate Pol I nuclease activity. As Pol I approaches the remaining ribonucleotides, it can remove them with its FEN domain following strand-displacement synthesis or nick translation. Synthesis and nucleotide removal continue until Pol I is outcompeted by DNA ligase, which repairs the final nick.

We interpret this result to mean that while a complete loss of 5′ nuclease activity from FEN and Pol I is lethal [35], maximal FEN activity is not necessary for normal growth. We conclude that *B. subtilis* cells grow well when some FEN activity is provided, and the existence of two pathways for Okazaki fragment maturation explains the lack of an obvious phenotype for the Δ*fenA* strain.

When we compare the activities of *B. subtilis* Pol I to Pol I from other organisms, we find some interesting differences. *B. subtilis* Pol I lacks 3′ to 5′ exonuclease [23] activity and we show here that Pol I FEN is weak. For organisms like *S. pneumoniae*, *H. influenzae* or *E. coli* that either lack a discrete FEN or have a discrete FEN with little to no measurable activity (in the case of *E. coli* Exo IX), the Pol I FEN domains show robust nuclease activity [29,70,71]. We suggest that when bacteria contain an active, discrete FEN, the 5′ nuclease activity of Pol I is no longer necessary, and Pol I could diverge to a more specialized role in Okazaki fragment resynthesis and DNA repair. In the case of organisms that lack an active, discrete FEN, these bacteria remain reliant on the FEN activity of Pol I. Therefore, we suggest that the differences in Pol I activity observed between organisms with a robust, discrete FEN may reflect the adaptation and specialization of each organism's genetic toolbox to manage the steps of Okazaki fragments and DNA repair more efficiently. For *E. coli*, *S. pneumoniae* and *H. influenzae* both tasks are completed by Pol I; for *B. subtilis* and the wide-spread group of bacteria that encode Pol I and FEN, we suggest that Okazaki fragments are matured using FEN and Pol I.

Our work shows that the major contribution of Pol I to the cell is through its DNA polymerase domain, for both Okazaki fragment maturation and DNA repair. Based on all available evidence, the contribution of FEN is specific to completion of Okazaki fragments. We propose that the combination of FEN and Pol I provide the most efficient process for primer removal and resynthesis during completion of the lagging strand. The two-protein system may provide *B. subtilis* with alternate pathways for lagging strand synthesis, which could be advantageous under conditions of genotoxic stress. Further, this mechanism, with separate proteins contributing FEN and DNA polymerase activity, is reminiscent of the mechanism of Okazaki fragment maturation employed by eukaryotic cells where FEN-1 (Rad27) and DNA polymerase δ coordinate during replication [73]. The maintenance of multiple members of the FEN family in the same organism can be seen beyond prokaryotes, as demonstrated by the FEN-1 and EXO1 proteins in eukaryotes [79]. As such, the differences between FEN and Pol I that we show here may support a broad pattern of specialized FENs in bacterial DNA replication and repair. Given the similarities we find between *B. subtilis* and eukaryotes, further study of lagging strand replication in *B. subtilis* may provide new mechanistic insights that will be relevant to Okazaki fragment processing in more complex systems.

## Methods

### Plasmid cloning

All primers used are listed in **S2 Table**. Plasmids were constructed using either pDR110 or pE-SUMO as the vector, which were amplified with oJR262 and 263 or oJR46 and 47, respectively. WT *B. subtilis* genes were amplified from PY79 genomic DNA, while FEN mutants were created using site-directed mutagenesis of plasmid carrying the WT gene. *polA_FEN* or *polA_Klenow* in the pDR110 vector were generated by selectively amplifying the appropriate regions from the *polA*-expressing plasmid. *xni* was amplified from *E. coli* BL21(DE3) cells. Genes were inserted into the appropriate vector using Gibson assembly, and subsequently used to transform *E. coli* MC1061 cells. Transformants with spectinomycin (pDR110 vector) or kanamycin (pE-SUMO vector) resistance were confirmed using colony PCR. Plasmid sequences were confirmed via Sanger sequencing.

### *B. subtilis* strain construction

All strains used in this work are described in **S3 Table**, and all *B. subtilis* strains are derivatives of PY79. Competent *B. subtilis* was generated by inoculating LB (supplemented with 3 mM $Mg_2SO_4$) with one colony of desired strain and grown in a rolling rack at 37°C until $A_{600nm}$ ~ 0.7. Pre-warmed MD minimal media (1x PC buffer [10x PC buffer: 107 g/L $K_2HPO_4$, 60 g/L $KH_2PO_4$, 10 g/L trisodium citrate • $(H_2O)_5$], 2% glucose, 50 μg/mL tryptophan, 50 μg/mL phenylalanine, 11 μg/mL ferric ammonium citrate, 2.5 μg/mL potassium aspartate, 3 mM $MgSO_4$) was inoculated with culture and grown in a rolling rack at 37°C for 6 hours. IPTG-inducible strains were constructed by transforming the appropriate parent strain with the NcoI linearized, pDR110-based plasmid containing the desired gene (plasmids are listed in **S4 Table**). Insertion at the *amyE* locus was confirmed by testing for a loss of starch-hydrolysis and through PCR amplification using oPEB866 and 867.

### Spot titers

*B. subtilis* strains were streaked onto appropriate LB plates and incubated overnight at 37°C. For each strain, 2 mL of LB was inoculated with a single colony and grown at 37°C in a rolling incubator to $A_{600nm}$ 0.5–1.0. Cultures were pelleted at 4000 x*g* and the supernatant replaced with sterile 0.85% w/v NaCl (saline). Each strain was normalized to $A_{600nm}$ = 0.5, then serially diluted to $10^{-5}$ in sterile saline. 5 μL of each dilution was spotted onto appropriate plates then incubated overnight. Plates were grown at 37°C due to cold sensitivity [13], unless noted to be grown at 30°C. All spot titers were performed in biological triplicate, and all plates were prepared on the day of the experiment. Brightness and contrast adjustment of whole images was performed using ImageJ.

### Protein expression

Competent BL21(DE3) were transformed with the appropriate plasmid (**S4 Table**), spread on LB agar supplemented with 25 μg/mL kanamycin, and incubated overnight at 37°C. A starter culture was made by inoculating LB containing 25 μg/mL kanamycin with a single colony, which was grown shaking at 200 rpm in a 37°C incubator overnight. A portion of the starter culture was used to inoculate each liter of LB supplemented with 25 μg/mL kanamycin. Cultures were grown at 37°C with shaking until $A_{600nm}$ ~ 0.6. Each liter of culture expressing FEN or FEN mutants was induced with 0.5 mM IPTG, cooled on ice for 20 min, and grown for 18 hours at 25°C. Cultures expressing Pol I were induced with 0.5 mM IPTG and grown at 37°C for 3 hours. Induced cells were harvested by centrifugation at 4°C and 4000 x*g* for 25 min. Supernatant was discarded and pellets were stored at -80°C.

### Protein purification

Two 1 L pellets were thawed on ice, then resuspended in Lysis Buffer (20 mM Tris, pH 8.0, 400 mM NaCl, 1 mM DTT). One protease inhibitor tablet (Pierce Protease Inhibitor Tablets, EDTA-free, A32965) was added, then cells were sonicated on ice for 2.5 minutes total on time (cycles of 10 seconds on and 20 seconds off) at 4°C. Lysed cells were centrifuged at 4°C and 14,000 x*g* for 45 minutes, then supernatant was decanted into a clean conical. Ni-NTA agarose (Qiagen, 30210) resin was equilibrated with Lysis Buffer in a gravity column, before clarified lysate was passed over it. The column was washed with 5 volumes of Lysis Buffer, followed by a wash with 10 column volumes of Wash Buffer (20 mM Tris, pH 8.0, 2 M NaCl, 1 mM DTT, 15 mM imidazole). Fractions containing protein were eluted with 3 column volumes of Elution Buffer (20 mM Tris, pH 8.0, 400 mM NaCl, 1 mM DTT, 300 mM imidazole), and the most

concentrated fractions were determined using a NanodropLite spectrophotometer. Selected fractions were pooled and treated with SUMO-protease for 2 hours at room temperature to remove the 6x-His-SUMO tag. Digested protein was transferred to 10K molecular cut off weight (MCOW) dialysis tubing and dialyzed overnight against Dialysis Buffer (20 mM Tris, pH 8.0, 300 mM NaCl, 1 mM DTT) at 4˚C overnight. The protein was passed over a Ni-NTA gravity column equilibrated with Dialysis Buffer, and the flowthrough was collected. The column was further washed with 3 column volumes each of Dialysis Buffer, Wash Buffer, and Elution Buffer. Each flowthrough was collected, and samples were analyzed using a 10% SDS-PAGE gel stained with Coomassie brilliant blue. For all proteins, the initial flowthrough and dialysis-wash flowthrough was pooled and concentrated using a 10K MCOW centrifugal filter unit (Amicon, UFC9010). Protein was stored in 25% glycerol at -80˚C. Further purification was completed using a HiTrap Q FF anion exchange column (Cytivia, 17515601) on an AKTA FPLC using Q Buffer A (20 mM Tris, 1 mM DTT, 5% glycerol) and Q Buffer B (20 mM Tris, 500 mM NaCl, 1 mM DTT, 5% glycerol). Briefly, the column was equilibrated with 10% Q Buffer B and protein was diluted to 50 mM NaCl in Q Buffer A before loading. Protein was fractionated using an increasing gradient of Q Buffer B. Peak fractions were determined using SDS-PAGE, then peak fractions were pooled and concentrated using a centrifugal filter. Glycerol was added to 25%, followed by flash freezing aliquots in liquid nitrogen before storing at -80˚C. Gel demonstrating protein purity was generated using by loading 1 µg of each protein onto a Mini-PROTEAN TGX 4–20% gradient gel (BIO-RAD, 4561096) and then staining with Coomassie blue.

## Nuclease activity assays

Substrates were generated by combining 1 µM labeled oligonucleotide with 2 µM of each appropriate unlabeled oligonucleotide in 1x Dilution Buffer (20 mM Tris, pH 8.0, 50 mM NaCl) and boiling for 1 minute at 98˚C before being allowed to cool to room temperature. The following oligonucleotides (synthesized by IDT) were used: nicked double-flap substrates contained oJR368, oFCL12 and either oJR339 or oJR340, gapped double-flap substrates used oJR368, oFCL10, and either oJR339 or oJR348, 5′ flap substrates included oJR366, oJR368, and either oJR339 or oJR348, nicked substrates contained oJR338, oJR340 and either oJR339 or oJR348, 3′ overhang structures consisted of oJR340 and either oJR339 or oJR348, blunt substrates consisted of oJR365 and either oJR339 or oJR348, 5′ overhang structures were oJR365 with oFCL4 or oFCL5, and single-stranded oligos were either oJR339 or oJR348. Sequences for all oligonucleotides are listed in **S1 Table**. Purified proteins were diluted to 250 nM in 1x Metals Buffer (20 mM Tris, pH 8.0, 50 nM NaCl, 1 mM $MgCl_2$, and 10 µM $MnCl_2$). Each reaction contained 100 nM substrate, 50 nM protein and 1x Metals Buffer. Assays were performed at 25˚C, and reactions were terminated with an equivalent volume of Stop Buffer (95% formamide, 20nM EDTA, bromophenol blue) at 0 s, 10 s, 1 min, and 15 min (unless otherwise noted in the figure caption). Stopped samples were incubated at 98˚C for 5 minutes, immediately followed by snap-cooling on ice. Each ladder was generated by incubating 500 nM of the chimeric form of a substrate in 200 nM NaOH at 37˚C for 8 minutes (5 minutes for the 5′ overhang), then stopped as described. Products were resolved using 20% denaturing urea-PAGE and visualized using the 800 nM channel of a LiCor Odyssey imager. Each reaction was repeated at least three times, using different preparations of substrate and different aliquots of protein. FEN and Pol I activity on the 5′ flap substrate in the presence of potassium (**S7 Fig**) was with the addition of 10 mM KCl to the 1x Metals Buffer. Our use of KCl was based on studies with *E. coli* ExoIX [41], although the referenced study used a higher KCl concentration.

## Analysis of nuclease activity assays

After a gel image was opened in ImageJ, the 0 s lane was selected and identified as the reference lane. The 10 s, 1 min, and 15 min lanes were also selected, and the lane intensities were plotted. Due to manufacturing limitations, the hybrid structure produces more background noise than the DNA-only substrates. The area under each curve was recorded using the Wand Tool, then measurements for each combination of protein/substrate/type were collected from three replicates and exported. Relative percent of uncut substrate was calculated by dividing the area for each timepoint by the area of the corresponding 0 s timepoint. Plots were made using ggplot2 [80] in R [81]; differences in protein activity on RNA-DNA hybrid substrate and DNA-only substrate were determined using Welch's T Test for unequal variance, an alpha of 0.05, and n = 3.

## Okazaki repair assays

Repair assays were performed using the method previously described [12,13]. Briefly, substrates were generated by combining 1 μM oJR367, 1 μM oFCL8, and 2 μM oFCL6 in 1x Dilution Buffer then boiling for 1 minute at 98°C and cooling to room temperature [9,13]. As noted in **S1 Table**, oFCL8 contains phosphorothioate bonds at the 5′ end to prevent off-target 5′ exonuclease activity. Reactions were carried out in 1x Extension Buffer (40 mM Tris-acetate, pH 7.8, 12 mM magnesium acetate, 300 mM potassium glutamate, 3 μM ZnSO4, 2% (w/v) polyethylene glycol, 0.02% pluronic F68), using 50 nM protein and 100 nM substrate, with or without 50 μM deoxynucleotide triphosphates (dNTPs). Assays were completed as described above for the Nuclease Activity Assays; however, an appropriate ladder was generated by combining the substrate (treated with sodium hydroxide) with an equal volume of 0.5 μM oFCL11 annealed to oFCL6. The dual-colored substrate was imaged using the 700 nM and 800 nM channels of a LiCor Odyssey imager.

## Supporting information

**S1 Fig. Strains with single deletions of *fenA* are not sensitive to UV damage.** Spot titer assay of the indicated strains exposed to UV damage.
(TIF)

**S2 Fig. The sequence of FEN contains eight carboxylate residues associated with bacterial FENs.** Multiple sequence alignment of Pol I (N-terminal FEN domain) and FEN from *B. subtilis* as well as the FEN homolog from *E. coli*, ExoIX. Conserved residues are boxed in grey while the active site carboxylate residues that coordinate metal-binding are boxed in pink.
(TIF)

**S3 Fig. FEN activity is dependent on conserved carboxylate residues.** Ectopic expression of *fenA* and *fenA* mutants in the Δ*rnhC*, Δ*fenA* strain grown following UV exposure.
(TIF)

**S4 Fig. Dominant negative phenotype of *fenA*[E114Q,D116N] and *fenA*[D189N,D192N] is not detectable in a WT background.** WT *B. subtilis* ectopically expressing *fenA* or *fenA* mutants with the indicated changes. Cells were imaged after growth at 30°C for 16 hours on the indicated concentration of hydroxyurea (HU).
(TIF)

**S5 Fig. *fenA*[E114Q,D116N] and *fenA*[D189N,D192N] are dominant negative in the absence of *rnhC*.** Overexpression of ectopic *fenA* or *fenA* mutants in cells lacking *rnhC* grown at different

concentrations of hydroxyurea.
(TIF)

**S6 Fig. Proteins used in assays were the major product of relative purifications.** A total of 1 μg of each protein for *in vitro* assays purified as described in the Methods section were electrophoresed on an SDS-PAGE. The gel was visualized following staining with Coomassie brilliant blue.
(TIF)

**S7 Fig. FEN and Pol I are not stimulated by the presence of potassium.** (A) Nuclease assays of FEN (B) FEN$^{D192N}$, and (C) Pol I on the simple flap substrate using a reaction buffer with 10 mM potassium (KCl). Substrates were generated using oligonucleotides oJR365 and either oJR339 (RNA-DNA hybrid; RNA indicated by zigzags) or oJR348 (DNA only), with a ladder produced via alkaline hydrolysis of the hybrid structure. Timepoints are as follows: 0 s, 10 s, 30 s, 1 min, 5 min, 15 min.
(TIF)

**S8 Fig. FEN and Pol I have different nuclease activities on 5′ overhang structures.** (A) Nuclease activity assays of FEN, (B) FEN$^{D192N}$, or (C) Pol I on duplex DNA with a 5′ overhang resolved using urea-PAGE. The mean percent substrate left intact by each protein was quantified from three replicates, shown underneath the respective assay. Standard deviation bars are provided, and statistical significance is indicated by asterisks as follows: * $p<0.05$, ** $p<0.01$, or *** $p<0.001$. (D) The FEN mutants FEN$^{E1414Q,D116N}$, FEN$^{Site1}$, and FEN$^{D189N,D192N}$ were also assayed for activity on the 5′ overhang structures. 5′ overhang structures were generated by annealing oJR365 with oFCL5 (RNA-DNA hybrid; RNA indicated by zigzags) or oFCL4 (DNA). Reaction time points are 0 s, 10 s, 1 min, and 15 min. Ladder was generated via alkaline hydrolysis of the hybrid 5′ overhang structure.
(TIF)

**S9 Fig. FEN has more nuclease activity on single-stranded substrates than Pol I.** (A) Activity of FEN, (B) FEN$^{D192N}$, or (C) Pol I on single-stranded RNA-DNA hybrid or DNA. The mean percent of intact substrate was quantified from three replicates and shown below the appropriate gel. Significance is indicated by asterisks (* $p<0.05$, ** $p<0.01$, or *** $p<0.001$) and standard deviation are provided. (D-F) FEN mutants were assayed on the same substrate, with representative gels shown from at least three replicates. Quantification of three replicates is shown below the associated gel. (G) Activity assay was repeated without the addition of protein (left) and percent substrate remaining intact was visualized graphically (right). Single-stranded RNA-DNA hybrid (RNA indicated by zigzags) was oligonucleotide oJR339 while single-stranded DNA was oJR348.
(TIF)

**S1 Table. Oligonucleotides used in *in vitro* assays.**
(DOCX)

**S2 Table. All primers used in this study.**
(DOCX)

**S3 Table. All strains used in this study.**
(DOCX)

**S4 Table. All plasmids used in this study.**
(DOCX)

**S1 Data. Numerical data underlying graphs.**
(XLSX)

## Acknowledgments

We would like to thank members of the Simmons lab for helpful discussions during the progression of this work.

## Author Contributions

**Conceptualization:** Frances Caroline Lowder, Lyle A. Simmons.

**Data curation:** Frances Caroline Lowder.

**Formal analysis:** Frances Caroline Lowder, Lyle A. Simmons.

**Funding acquisition:** Lyle A. Simmons.

**Investigation:** Frances Caroline Lowder, Lyle A. Simmons.

**Methodology:** Frances Caroline Lowder.

**Project administration:** Lyle A. Simmons.

**Resources:** Frances Caroline Lowder.

**Supervision:** Lyle A. Simmons.

**Validation:** Frances Caroline Lowder, Lyle A. Simmons.

**Visualization:** Frances Caroline Lowder, Lyle A. Simmons.

**Writing – original draft:** Frances Caroline Lowder, Lyle A. Simmons.

**Writing – review & editing:** Frances Caroline Lowder, Lyle A. Simmons.

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
