## [Decision Letter · Decision Letter 0]

31 Jan 2023

Dear Dr Simmons,

Thank you very much for submitting your Research Article entitled 'Bacillus subtilis  encodes a discrete flap endonuclease that cleaves 

RNA-DNA hybrids' to PLOS Genetics.

The manuscript was fully evaluated at the editorial level and by independent peer reviewers. The reviewers appreciated the attention to an important topic but identified some concerns that we ask you address in a revised manuscript.

Reviewers #1 and #2 consider that text and figures can be improved by some corrections, by adding additional information and by discussing some data that are omitted from being discussed.

Reviewer #3 considers that the conclusion that FEN rather than PolI-fen domain furnishes the principal Okazaki-fragment-processing flap endonuclease may be flawed because of several issues concerning materials used and experimental procedure.

We therefore ask you to modify the manuscript according to the review recommendations. Your revisions should address the specific points made by each reviewer.

Yours sincerely,

Ivan Matic

Academic Editor

PLOS Genetics

Lotte Søgaard-Andersen

Section Editor

PLOS Genetics

Reviewer's Responses to Questions

**Comments to the Authors:**

Reviewer #1: Summary

The authors use Bacillus subtilis as a model to investigate the role of a flap endonuclease (FEN) protein in removal of RNA primers in Okazaki fragment maturation. They use genetic and biochemical approaches to do this which mainly take the form of endonuclease assays on model oligonucleotide substrate. The authors found that the B. subtilis protein FEN is involved in Okazaki fragment maturation through growth assays done under DNA damaging conditions and observed increased sensitivity to DNA damage in the absence of both FEN and ribonuclease HIII. To understand FEN’s activity in Okazaki fragment maturation, they purified the protein and measured endonuclease activity on a variety of substrates that mimic RNA-DNA hybrids that form during Okazaki fragment synthesis and those that are structurally similar but are exclusively constructed from DNA. They determined that FEN has increased activity on substrates that contain portions that are RNA compared to DNA-only substrates. They also observed increased activity on structures mimicking Okazaki fragments compared to structures not expected in this pathway.

Critique

Okazaki fragment maturation is an important biological problem and B. subtilis is a good model system in which to study this. Additionally, FENs are ubiquitous in biology, so the activities learned in this system may shed light on other DNA metabolic pathways and pathways in other organisms. The work in this manuscript is very interesting and experiments appear to be well controlled. On the whole, the text is very straightforward and easy to read. One criticism, though, is that at times in the interest of conciseness and brevity, some additional nuance and analysis of the data are omitted from being discussed. This is detailed below in addition to other points.

• Although an elegant model, the impact of this work suffers a bit because of the similarities between the proposed pathway and pathways already established in eukaryotes. I don’t think that this undermines the importance of the work, but the authors should add more in the discussion about this. E.g.--because they are similar, and because bacterial models are easier than eukaryotic, more detailed mechanistic work could be done using the B. subtilis system that will elucidate not just bacterial pathways but also eukaryotic.

• It’s interesting that the fenA-D192A rescues the double deletion phenotype in Figure 1D and has reduced catalytic activity in to the biochemical experiments. The authors mention that previously this mutant was suggested to be catalytically inactive when clearly the authors show activity. It would be helpful to add a brief explanation for this discrepancy.

• The data in Figure 1 is clear, but it would strengthen the author’s argument for a role for FEN RNA primer removal and Okazaki fragment maturation if there is also a phenotype with deletion of another protein in the pathway (can ligase be used?). This may not be possible, but an alternative may be to show a lack of phenotype with a double deletion of fenA and a RNase not predicted to play a role in Okazaki fragment processing as a control.

• It would be helpful in the endonuclease assays, if the ladder, gel, and substrate cartoons could be labeled with sizes of the cleavage products. It appears that FEN is cleaving at a discrete position that is inferred to be at the RNA-DNA segment junction point on the strand. It would be useful to show that.

• Relatedly, it would be helpful if the authors could provide additional commentary and discussion about their observed activities of FEN. For instance, it is clearly cleaving the DNA at a specific site (there is a single product band and opposed to a ladder or smear). This suggests a specific substrate recognition ability. Additionally, quantitatively comparing cleavage rates for each substrate and protein/variant would strengthen the author’s presentation of data. The authors’ data is quite clear and seems to contain more mechanistic observations where additional analysis would be interesting.

• I am confused about the disappearance of the green labeled primer strand in lane 6 in Figure 8. Does it get extended, but to a variety of lengths, so it does not resolve as a single band?

• Also, in this figure, what is the high molecular weight product in lane 7? Does Pol I extend beyond the template strand? If so, how?

• It would also strengthen the model, if the authors could use this assay to partially reconstitute. If they add both Pol I and FEN can they observe primer extension and RNA flap removal? They may need a nuclease deficient Pol I to do this.

Reviewer #2: Lowder and Simmons examine the activity and roles of B. subtilis FEN (aka YpcP or ExoR) in vitro and in vivo. Their results demonstrate a role for FEN in DNA repair in vivo when RNase HIII is absent and suggest a role for FEN in processing RNA primer flap structures in Okazaki fragment processing in vivo. A systematic biochemical study reveals robust nuclease activity for FEN on a variety of substrates, with a preference for RNA:DNA hybrids and 5’ DNA flaps. Purified FEN appears to be more active as a nuclease than DNA polymerase I on the same substrates, consistent with FEN’s role in Okazaki fragment processing. The manuscript is very well written and the conclusions are well supported. The findings represent a significant advancement in our understanding of how FEN participates in DNA replication, suggesting an unexpected similarity between lagging strand processing in B. subtilis and eukaryotic systems.

Concerns:

(1) There appear to be some problems with Figure 1. In Figures 1B and C, D192 is not labeled. Instead, it appears that residue 192 is mislabeled as D190. A second issue is in Figure 1D. The authors state on page 8 that overexpression of the fenA D192N mutant rescued HU sensitivity whereas the fenA D189N, D192N mutant did not. However, the results are the opposite in Figure 1D, which makes me think that the labels are switched on the two mutant strains. The authors should confirm that these are correct.

(2) Metal concentrations in enzymatic assays include 10 mM MgCl(2) and 10 uM MnCl(2), which the authors note reflect the expected physiological concentrations. The reference that supports this statement is a prior paper from the Simmons group where RNases H from B. subtilis are tested in vitro in differing divalent metal concentrations. A reference to a paper that measured free divalent metal concentrations in B. subtilis would be more appropriate.

(3) Given the complex relationship of metal-dependent nucleases on divalent metal identity, it would be nice to know if the authors have examined FEN activity in MgCl(2) or MnCl(2) alone to determine whether activity requires both. This is not essential for this paper, but a study similar to ref. 12 could be a useful contribution to the literature.

(4) In figure 8, why is the green product observed with Pol I and dNTPs longer than the template strand?

(5) In figure 10, it appears that the strain overexpressing E. coli xni is the most sensitive to HU treatment – why would this be? Could ExoIX be acting in a dominant-negative manner? Since ExoIX can bind to SSB in E. coli, perhaps overexpressed ExoIX is occupying binding sites on SsbA and blocking proper complex formation with other SsbA binding partners?

Reviewer #3: Substrate forms an odd structure not quite the same as the classical “double-flap (5’ flap with 3’ single nucleotide flap” preferred by several reported FENs e.g. yeast (https://doi.org/10.1074/jbc.M606582200); https://doi.org/10.1074/jbc.M110662200; Archaeoglobus fulgidus https://doi.org/10.1016/S0092-8674(03)01036-5 ; human, doi: 10.1038/ncomms15855 , https://doi.org/10.1074/jbc.M115.666438; bacteria https://doi.org/10.7554/eLife.62046

All organisms from bacteria to humans and even some viruses appear to require a FEN activity for replication. These metalloproteins require divalent metals such as magnesium or manganese for activity. Many bacteria possess two distinct FEN homologues which is the case in B. subtilis and this manuscript examines a pair of FEN homologues. In contrast to human FEN1 and its eukaryotic homologues, bacterial fenA gene products have received significantly less attention than other family members, with most studies concentrating on the DNAPolI-FEN domain. FENs are generally accepted as being required for Okazaki fragment processing and the observation that many bacteria contain tow paralogues is intriguing as for example Rosenberg and coworkers were unable to identify any phenotype associated with the E. coli discrete FEN hiomologue ExoIX.

In the current manuscript, the biochemical characterisation of the fenA gene product, a discrete FEN protein is compared with that of DNA polymerase I’s FEN domain on a number of substrates. It also reports on a number of genetic experiments attempting to unravel the interdependencies of RNase H, FEN and DNA Pol1. These combined comparative biochemical an genetic studies lead the authors to conclude that the fenA gene product (FEN) is mainly responsible for removing the RNA portion of maturing Okazaki fragments rather than the polymerase’s own FEN-domain (located on the N-terminal 1/3rd of the protein, the C-terminal carrying the polymerase and proofreading domains, equivalent to the Klenow, or large fragment in E. coli). One of their results suggests that the DNA PolI fen domain cannot rescue a deletion of the entire PolI gene. This is in stark contrast with the situation in e.g. Strep. Pneumoniae or Haemophilus influenzae and may be quite intriguing.

The manuscript itself is mostly very well written, quite comprehensive and sets the scene for the study perfectly. The results section contains 11 clearly presented figures plus additional supplementary information showing the biochemical activity the FEN protein, various mutants designed to reduce or ablate activity by altering the active site carboxylate residues required for divalent-metal ion binding.

The conclusion that the fenA gene product has exo- and endo-nuclease activities appears reasonable as is the interpretation of the requirement for divalent metal ions. The knock-out and complementation studies, etc are also clearly described and largely well presented.

However, the conclusion that FEN rather than PolI-fen domain furnishes the principal Okazaki-fragment-processing flap endonuclease is, in this reviewer’s opinion potentially flawed for the reason given below (Major Point 1).

MAJOR POINT

1) The main substrates used for the in vitro work described do not reflect the generally accepted structure of a FEN substrate as implied in the text (lines 199-200). A FEN substrate should consist of a so-called double flap. That is a 3’ single-nucleotide “flap” or invader, adjacent to the 5’-flap on the template. The 3’ flap available to base pair with the template-strand nucleotide revealed when the last ribonucleotide is removed upon 5’-flap excision. This generates a double-stranded substrate with a nick suitable for DNA ligase. However, when I drew out the structure of the substrates used (Fig2, A and elsewhere) this is not the case. The product of a 12 nucleotide (the RNA region) cleavage yields a ds-DNA with a 4 base gap between the 3’ end of oJR366 and DNA-only product of cleavage of oJR339 (last 9 nucleotides ATGCTTACG), when annealed to the template oJR368. This is not a classical double-flap. Several reports on the preferred classical FEN substrate concur that they consist of the “double-flap” e.g. yeast (https://doi.org/10.1074/jbc.M606582200); https://doi.org/10.1074/jbc.M110662200; Archaeoglobus fulgidus https://doi.org/10.1016/S0092-8674(03)01036-5 ; human, doi: 10.1038/ncomms15855 , https://doi.org/10.1074/jbc.M115.666438; bacteria https://doi.org/10.7554/eLife.62046; this list is not exhaustive. The substrate used has 4 bases of complementary RNA-to template DNA (9-12), thus when cleaved to the product 12 mer produce a gapped substrate unless there has been a mistake with the sequences in the paper.

2) The very low level of detectable flap endonuclease activity exhibited in the B. subtilis DNA PolI holoenzyme is somewhat hard to understand. Have the conditions used been explored or optimised for PolI? It is noteworthy that the conditions used in the “nick translation” (Okazaki fragment repair) figure (8) are very different from those used in the previous nuclease characterisation figures. Why is this?

3) Nuclease activity comparisons in Figs. 2 to 7 appear to have been carried out using buffers lacking an appreciable concentration of potassium and at very high enzyme concentrations compared with DNA (100 nanoM and 50 nanoM, respectively). Earlier work has indicated that several, but not all, FENs are greatly stimulated by the presence of a potassium ion bound in the helix-2(3)-turn helix region of FEN proteins from bacteria to man.

4) The experiments in Figs. 2-7 using high concentrations of FEN protein, were essentially complete in 1 min for the WT reaction – the product which appears to be derived by endonuclease could also have been derived by a processive EXO-nuclease action. Indeed, in Fig. 2B, the “slower” mutant shows clear evidence of bands longer than the final 12 mer product at 1 min, they seem to be 13- 15 nucleotides, with no final 12 mer until 15 min. Indeed, even in Fig 2A, there is a hint of intermediate products in 10 s and 1 min lanes. Fig 3A similarly, shows roughly equal amounts of 13 and 12 mer product in the 10s lane, some 13mer but mostly 12 mer at 1 min and all 12 mer at 15 min.

5) Lines 310 – expression of Pol1FEN did not reduce sensitivity… fig. 9 . I may be misinterpreting this but the figure as shown, suggest the FEN domain complemented strain (lower right hand panel) shows growth much more growth at 10E-3 and 10E-4 than the Klenow complemented ones (no colonies past 10E-2)?

Minor issues

6) Line 188-189- references materials and methods. There is no M&M section, rather just Methods.

7) Several uses of rpm for centrifuge speeds (e.g. line 453 and elsewhere) , better to give x g (note g should be in italic).

8) OD (line 450 and elsewhere) is a defunct term, better to use A600 or A660.

9) While the paper is largely thoughtfully and comprehensively referenced, the authors might wish to include comparison of their PolFEN-domain complementation with the situation in Haemophilus and Streptococcus.

10) References contain a mix of title and sentence case.

**Have all data underlying the figures and results presented in the manuscript been provided?**

Reviewer #1: Yes

Reviewer #2: Yes

Reviewer #3: Yes

PLOS authors have the option to publish the peer review history of their article (what does this mean?). If published, this will include your full peer review and any attached files.

Reviewer #1: No

Reviewer #2: No

Reviewer #3: No

---

## [Decision Letter · Decision Letter 1]

11 Apr 2023

Dear Dr Simmons,

Thank you very much for submitting your Research Article entitled 'Bacillus subtilis  encodes a discrete flap endonuclease that cleaves RNA-DNA hybrids' to PLOS Genetics.

The manuscript was fully evaluated at the editorial level and by independent peer reviewers. All three reviewers recommended your article for publication. However, Reviewer #3 has suggested a few minor modifications.

Therefore, before final acceptance, we request that you modify the manuscript according to the recommendations made by Reviewer #3

Yours sincerely,

Ivan Matic

Academic Editor

PLOS Genetics

Lotte Søgaard-Andersen

Section Editor

PLOS Genetics

Reviewer's Responses to Questions

**Comments to the Authors:**

Reviewer #1: The authors have done a nice job responding to reviewer comments and the paper is much clearer and improved. I have no concerns that warrant additional revision and recommend publication.

Reviewer #2: I am satisfied with the authors revisions and am pleased to accept the paper for publication as is.

Reviewer #3: The manuscript has been revised appropriately. One minor comment is that in line 580 they state that they carried out assays in the presence of "10 mM KCl according to ref. 41."

The quoted reference employed 100 mM KCl in ExoIX reactions (page 8359, of ref. 41, "Reactions were carried out using 50 pM annealed flap substrate in 25 mM potassium glycinate (pH 9.3), 100 mM KCl, 1 mM DTT, 0.5 mM EDTA, with or without 10 mM MgCl2."

The authors may wish to add a caveat that any differences observed maybe due to the lower KCl concentration used.

Minor issues:

1) Line 26- "5’ ssRNA" an apostrophe is incorrectly used - this should be 5 prime (symbol).

2) Line 81 and elsewhere. The term bivalent cation is used. Divalent seems to be more widely used in the literature.

3) Line 85, delete "small" it is redundant as "single nucleotide" is specified.

4) Line 271, "preference opposite that of FEN" should read " preference opposite to that shown by FEN."

5) Line 314/5 "like what is seen here with PolI" Should read "as seen here with PolI"

6) Line 508 and elsewhere "A600" should be A (subscript 600nm)?

7) Line 516 "BL21-subscript-DE3-(FCL14)" Usually BL21 with the DE3 prophage is written as BL21(DE3), not subscript? Is FCL14 a supplier or is this a variant of BL21(DE3) ? Similarly, line 599.

8) Line 549 and elsewhere - "(20mM Tris, 500mM NaCl, 1mM DTT") - spaces between numbers and units missing should be "20 mM Tris, 50 0mM NaCl, 1 mM DTT".

9) Fig 1- Please consider adding in a panel showing a picture of the entire protein, rather than just the active site. This will provide context for readers who are unfamiliar with FEN architecture.

10) On page 34, legend to Sup Fig. 7, KCL is used, the "L" should be lower case- KCl.

**Have all data underlying the figures and results presented in the manuscript been provided?**

Reviewer #1: Yes

Reviewer #2: Yes

Reviewer #3: Yes

PLOS authors have the option to publish the peer review history of their article (what does this mean?). If published, this will include your full peer review and any attached files.

Reviewer #1: No

Reviewer #2: No

Reviewer #3: No

---

## [Editor Report · Decision Letter 2]

18 Apr 2023

Dear Dr Simmons,

We are pleased to inform you that your manuscript entitled "Bacillus subtilis  encodes a discrete flap endonuclease that cleaves RNA-DNA hybrids" has been editorially accepted for publication in PLOS Genetics. Congratulations!

Yours sincerely,

Ivan Matic

Academic Editor

PLOS Genetics

Lotte Søgaard-Andersen

Section Editor

PLOS Genetics

Comments from the reviewers (if applicable):

**Data Deposition**

http://datadryad.org/submit?journalID=pgenetics&manu=PGENETICS-D-22-01454R2

**Press Queries**

---

## [Editor Report · Acceptance letter]

2 May 2023

PGENETICS-D-22-01454R2 

*Bacillus subtilis* encodes a discrete flap endonuclease that cleaves RNA-DNA hybrids 

Dear Dr Simmons, 

We are pleased to inform you that your manuscript entitled "*Bacillus subtilis* encodes a discrete flap endonuclease that cleaves RNA-DNA hybrids" has been formally accepted for publication in PLOS Genetics! Your manuscript is now with our production department and you will be notified of the publication date in due course.

With kind regards,

Anita Estes

PLOS Genetics

On behalf of:
